# Development of a genetically encoded sensor for probing endogenous nociceptin opioid peptide release

Xuehan Zhou [1,2,13], Carrie Stine [3,4,5,13], Patricia Oliveira Prada [6,7,8], Debora Fusca [7,9], Kevin Assoumou [10], Jan Dernic [1], Musadiq A. Bhat [1], Ananya S. Achanta [3,4], Joseph C. Johnson [3,4], Amanda Loren Pasqualini [3,4], Sanjana Jadhav [3,4], Corinna A. Bauder [6,7], Lukas Steuernagel [6,7], Luca Ravotto [1], Dietmar Benke [1,2], Bruno Weber [1,2], Azra Suko [3,4], Richard D. Palmiter [3,11], Miriam Stoeber [10], Peter Kloppenburg [7,9], Jens C. Brüning [6,7,12], Michael R. Bruchas [3,4,5] ✉ & Tommaso Patriarchi [1,2] ✉

Nociceptin/orphanin-FQ (N/OFQ) is a recently appreciated critical opioid peptide with key regulatory functions in several central behavioral processes including motivation, stress, feeding, and sleep. The functional relevance of N/OFQ action in the mammalian brain remains unclear due to a lack of high-resolution approaches to detect this neuropeptide with appropriate spatial and temporal resolution. Here we develop and characterize NOPLight, a genetically encoded sensor that sensitively reports changes in endogenous N/OFQ release. We characterized the affinity, pharmacological profile, spectral properties, kinetics, ligand selectivity, and potential interaction with intracellular signal transducers of NOPLight in vitro. Its functionality was established in acute brain slices by exogeneous N/OFQ application and chemogenetic induction of endogenous N/OFQ release from PNOC neurons. In vivo studies with fibre photometry enabled direct recording of NOPLight binding to exogenous N/OFQ receptor ligands, as well as detection of endogenous N/OFQ release within the paranigral ventral tegmental area (pnVTA) during natural behaviors and chemogenetic activation of PNOC neurons. In summary, we show here that NOPLight can be used to detect N/OFQ opioid peptide signal dynamics in tissue and freely behaving animals.

The nociceptin/orphanin-FQ peptide (N/OFQ), along with its cognate receptor (NOPR) encoded by *Oprl1* gene, represents the most recently discovered opioid peptide/receptor system[1,2]. NOPR is a G protein-coupled receptor (GPCR) which shares 60% of sequence similarity with the other members in the opioid family[3], while retaining a unique pharmacological profile[4]. Upon occupancy by its endogenous peptide ligand N/OFQ, the receptor activates downstream Gi/Go proteins and induces intracellular signaling that includes the inhibition of cAMP formation, and ultimately reduces neurotransmission via the inhibition

of voltage-gated calcium channels and the activation of inwardly-rectifying potassium channels[5].

NOPR is abundantly expressed within the central nervous system[6–8], in line with the broad range of neural and cognitive functions regulated by this endogenous opioid system[9,10]. In particular, NOPR and preproN/OFQ (PNOC)-expressing neurons are highly enriched in the ventral tegmental area (VTA), arcuate nucleus of the hypothalamus (ARC), dorsal striatum, nucleus accumbens (NAc), and medial prefrontal cortex (mPFC)[7]. Being the major source of dopamine

to limbic and forebrain regions, the VTA plays an important part in neural circuits regulating motivation and reward-based learning[11,12]. It is well established that VTA dopaminergic projections to the NAc are essential for encoding reward prediction error and adaptive motivated behavior towards both beneficial and aversive stimuli[13]. Several studies have shown that the N/OFQ system exerts an important modulatory effect on mesolimbic dopaminergic circuits. For example, intracerebroventricular (ICV) injection of N/OFQ produces a decrease in extracellular dopamine in the NAc[14] and exerts an inhibitory constraint on dopamine transmission by either inhibiting tyrosine hydroxylase phosphorylation or dopamine D1 receptor signaling[15]. At the behavioral level, N/OFQ has been shown to prevent morphine- and cocaine-induced dopamine increase in the NAc[16,17], and inhibit conditioned place preference to morphine, amphetamine, and cocaine[18,19], while disruption of the N/OFQ system is associated with motivated responding disorder[20]. ICV administration of N/OFQ was reported to potently block reward-associated cues but showed no effect on aversion-associated cues[21]. In a recent study, we identified a subgroup of PNOC-enriched neurons located in the paranigral VTA which, when activated, caused avoidance behavior and decreased motivation for reward[22]. Additionally, a separate population of PNOC neurons located in the ARC has emerged as an important neuronal population involved in regulating feeding behavior. These GABA-expressing neurons are activated after three days of a high-palatable, energy-dense diet and have been found to play a crucial role in feeding control. In particular, optogenetic stimulation of PNOC-expressing neurons in the ARC induces feeding, while selective ablation of these neurons decreases food intake and prevents obesity[23]. Apart from its roles in the VTA-NAc and ARC circuits, N/OFQ can also inhibit mPFC-projecting VTA neurons[24] and a reduction in mPFC N/OFQ level was reported in rodents that underwent conditioned opioid withdrawal[25].

Of note, past studies on N/OFQ signaling have generated some contradictory results[26]. Activation of the NOPR with a selective agonist was reported to reduce alcohol drinking and seeking behavior[27,28], while a selective NOPR antagonist, LY2940094, was reported to have the same effect[29]. In anxiety-related behaviors, it has been reported that central injection of a NOPR agonist induces anxiogenic effect, but anxiolytic effects of NOPR agonists had also been reported[30–32]. These observations can be interpreted in different ways. The dynamics of NOPR desensitization after the application of agonists or antagonists, for example, could contribute to these contradictory results. Another possible explanation could be the competition of different local neural circuits simultaneously recruited by the N/OFQ system, as most of the studies mentioned before do not have fine spatial control over the application of drugs nor the resolution to isolate endogenous release dynamics of the peptide. Overall, the exact mechanism of the NOPR-N/OFQ system and its impact on different neural circuits are, at best, only partially understood.

A major factor hindering a clearer understanding of N/OFQ regulation of neural circuits, or any neuropeptide signaling system, is the limitations imposed by current tools and techniques used to detect the release of neuropeptides in living systems. Conventional techniques such as microdialysis and mass spectrometry-coupled high-performance liquid chromatography can successfully detect picomolar levels of neuropeptides in extracellular fluid[33], yet the spatial and temporal resolution of these techniques is limited to single point measurements and long timescales on the order of minutes[34]. This temporal and spatial resolution is low for decoding the neuronal mechanisms of dynamic peptide action in vivo, particularly alongside other neurophysiological methods. New approaches to probe the nervous system using fluorescent sensors have started to gain traction across the field[35–39]. Combined with rapidly developing fluorescent recording and imaging techniques, these sensor-based approaches are uniquely suited for in vivo observations with finer spatiotemporal resolution than was previously possible[35,40,41].

Here we report the development and characterization of NOP-Light, a genetically encoded opioid peptide sensor that provides a specific and sensitive fluorescence readout of endogenous N/OFQ dynamics with high temporal resolution ex vivo and in vivo. Using the sensor, we could detect ligand binding by systemically administered NOPR agonists and antagonists within the central nervous system. We also measure both chemogenetically evoked and behaviorally induced dynamics of endogenous N/OFQ in freely moving mice. Thus, NOP-Light extends the neuropeptide molecular toolbox necessary to investigate the physiology of neuropeptides and, in particular, this important endogenous opioid system with high resolution.

## Results
### Development of a genetically encoded N/OFQ opioid peptide (N/OFQ) sensor
To develop a fluorescent sensor for N/OFQ, we started by designing a prototype sensor based on the human NOPR which has 93–94% sequence identity to the mouse and rat receptors. We replaced the third intracellular loop (ICL3) of human NOPR with a circularly-permuted green fluorescent protein (cpGFP) module that was previously optimized during the development of the dLight1 family of dopamine sensors[41] (Fig. 1a and Supplementary Fig. 1a). This initial construct exhibited poor membrane expression and no fluorescent response to N/OFQ (Supplementary Fig. 1b). Given the pivotal role of the GPCR C-terminus in trafficking[42], we reasoned that replacement of the NOPR C-terminus with that of another opioid receptor may facilitate membrane targeting of the sensor. Based on our prior experience[41], we chose to use the C-terminus from the kappa-type opioid receptor. The resulting chimeric receptor showed improved expression at the cell surface, but still exhibited only a small response to N/OFQ (Supplementary Fig. 1b). We then aimed to improve the dynamic range of the sensor through mutagenesis efforts. First, we elongated the N-terminal cpGFP linker with additional amino acids originating from dLight1[41]. This led to the identification of a variant with a fluorescent response ($\Delta F/F_0$) of ~100% (Supplementary Fig. 1c, d). Prior work demonstrated that sequence variations in the second intracellular loop (ICL2) of the sensor can effectively be used to modulate sensor response[41,43]. Thus, as a next step, we performed targeted mutagenesis focusing on the ICL2 of the sensor. Through these efforts we identified a beneficial mutation (I156[34,51]K, Supplementary Fig. 1e) that was then carried forward onto the next rounds of screening, which focused on receptor and cpGFP residues around the insertion site of the fluorescent protein between transmembrane helixes 5/6 (TM5/TM6) (Supplementary Fig. 1e). The final variant, which we named NOPLight, had a $\Delta F/F_0$ of 388% in transfected HEK293T cells and a similar performance in transduced neuron culture ($\Delta F/F_0 = 378\%$) upon activation by the high affinity, full agonist N/OFQ (Fig. 1b). Furthermore, the evoked fluorescence signal could be reversed to baseline levels using the selective and competitive small-molecule NOPR antagonist J-113397 (Fig. 1c). To aid the subsequent characterization experiments we also developed a control sensor, NOPLight-ctr, by mutating into alanine two key residues (D110[2.63], D130[3.32]) located in the binding pocket of NOPR[4] (Supplementary Fig. 2a and Supplementary Note S1). The control sensor was well expressed on the surface of HEK293T cells and neurons but showed negligible fluorescent response to N/OFQ (Fig. 1c, d and Supplementary Fig. 2a) and a panel of other endogenous opioids and fast neurotransmitters, including dopamine, acetylcholine, and GABA. (Supplementary Fig. 2b).

### In vitro characterization of NOPLight
To better examine the pharmacological and kinetic properties of the sensor, we first characterized the apparent ligand affinity of NOPLight in vitro, using NOPLight-expressing HEK293T cells and cultured neurons. In HEK293T cells, the endogenous ligand N/OFQ elicited a

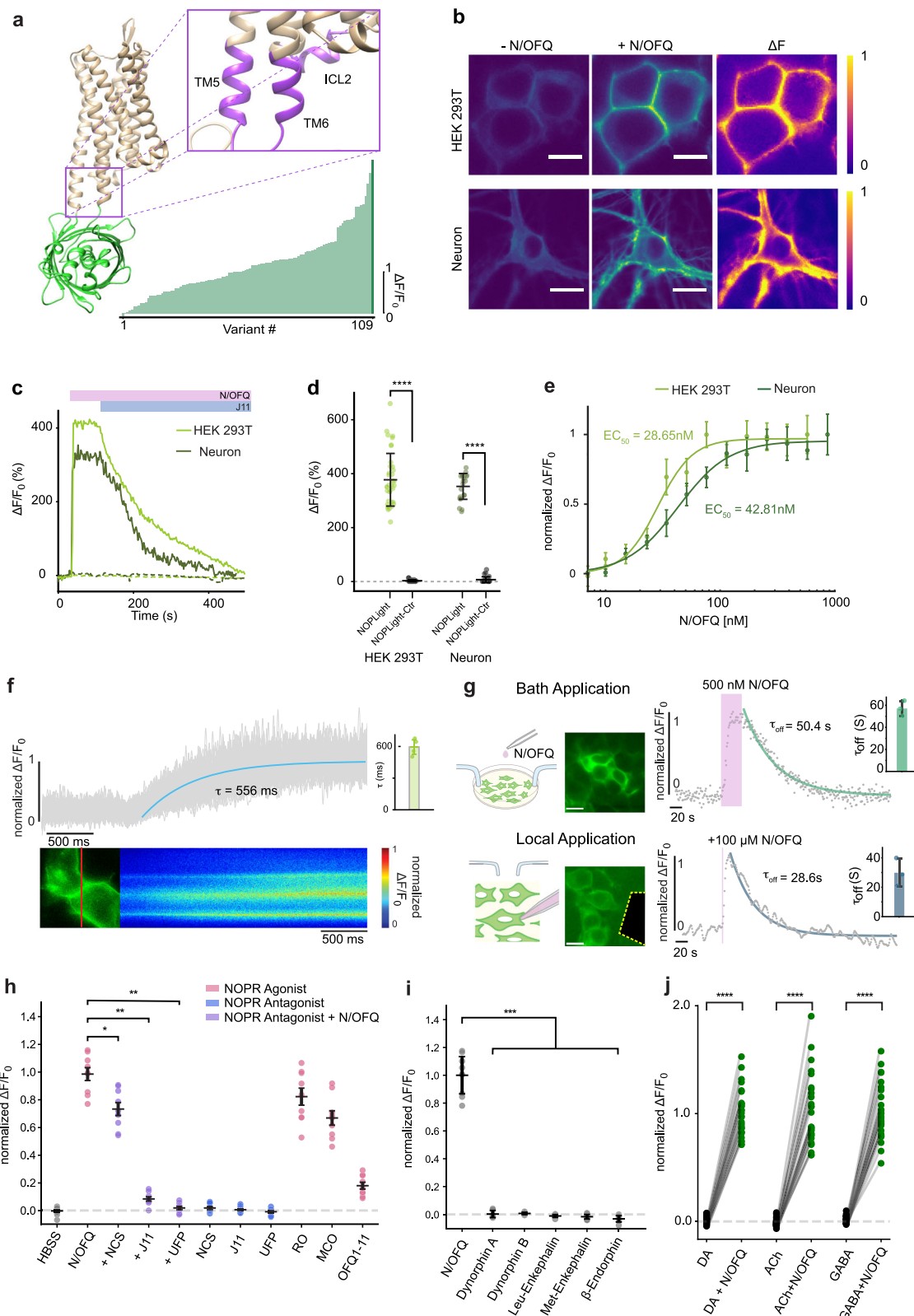

fluorescent response of NOPLight at a half maximal effective concentration ($EC_{50}$) of $28.65 \pm 5.1$ nM ($pEC_{50} = 7.54$), whereas in cultured neurons it showed an $EC_{50}$ of $42.81 \pm 5.4$ nM ($pEC_{50} = 7.37$) (Fig. 1e), approximately one order of magnitude lower than the reported potency of N/OFQ to the wild-type NOPR in the central nervous system[44]. To determine the activation kinetics of NOPLight, we measured the activation of NOPLight upon direct bath application of N/

OFQ using high-speed line-scan confocal imaging. Mono-exponential fitting of NOPLight fluorescent response indicated a subsecond time constant of signal activation at the sensor ($\tau_{ON} = 595 \pm 69$ ms; Fig. 1f). To determine the off kinetics of the sensor, we performed experiments on cells under constant bath-perfusion. Depending on the experimental set-up, the fluorescence response of NOPLight elicited by N/OFQ application returned to baseline with a $\tau_{off}$ of $57.1 \pm 6.9$ s for bath

**Fig. 1 | Development and in vitro characterization of NOPLight. a** Structural model of NOPLight and screening summary. $n \geq 3$ cells for each variant. **b** Representative images of HEK293T cells (top; scale bars, 10 μm) and neurons (bottom; scale bars, 20 μm) expressing NOPLight before and after application of 1 μM N/OFQ and ΔF heat map. **c** Responses of NOPLight or NOPLight-ctr in HEK293T cell and neurons to 1 μM N/OFQ, followed by competition with 10 μM J-113397. **d** Quantification of maximal responses from (**c**), $n = 38, 30, 22, 27$ cells from three independent experiments, respectively. Data shown as mean ± SD. (One-sided Mann–Whitney U-test, ****$P < 0.0001$, $P = 1.003 \times 10^{-12}$, and $1.262 \times 10^{-9}$ for NOPLight versus NOPLight-ctr in HEK293T and neurons). **e** Normalized response of HEK293T cells and neurons expressing NOPLight to N/OFQ titrations (mean ± SD) fitted with three-parameter Hill equation. $n = 3$ independent experiments per concentration. **f** Line-scan time plot of NOPLight $\Delta F/F_0$ (gray) from membrane-pixels shown in bottom. Right, quantification of time constant (τ) from four independent experiments. Data represented as mean ± SD. **g** Left: experimental

schematics and images (scale bars, 20 μm), right: time plot of normalized $\Delta F/F_0$ from a representative experiment (gray) with quantified time constant ($\tau_{off}$). Data represented as mean ± SD. **h** Maximal NOPLight response to: NCS, Nocistatin; J11, J-113397; UFP, UFP-101; RO, Ro 64-6198; MCO, MCOPPB; OFQ1-11, Orphanin-FQ (1–11)). *$P < 0.05$, **$P < 0.01$, $P = 0.024$, Nocistatin; 0.0012, J-113397; 0.0012, UFP-101. $n = 9$ cells except MCO ($n = 8$) from three independent experiments. **i** Maximal NOPLight response to endogenous opioid ligands (1 μM). **$P < 0.01$, $P = 0.003$, Dynorphin A; $P = 0.003$, Dynorphin-B; $P = 0.003$, Leu-Enkephalin; $P = 0.003$, Met-Enkephalin; $P = 0.003$, β-endorphin. $n = 9$ from three independent experiments. Statistical comparisons in (**h**, **i**) by one-sided pairwise Mann–Whitney rank test with post hoc Bonferroni correction, data represented as mean ± SEM. **j** NOPLight response to neurotransmitters (DA dopamine, ACh acetylcholine, GABA gamma-Aminobutyric, all 1 mM). $n = 29$ (DA) or 30 (ACh, GABA) cells from three independent experiments (two-sided paired t-test, ****$P < 0.0001$, $P = 2.45 \times 10^{-22}$, $7.46 \times 10^{-19}$, and $7.82 \times 10^{-21}$, respectively).

application and $29.9 \pm 9.5$ s for localized puff application of the ligand (Fig. 1g).

We characterized the pharmacological profile of NOPLight in vitro. We tested the response of NOPLight-expressing HEK293T cells to a panel of small-molecule and peptide ligands that are known as NOPR agonists or antagonists (Fig. 1h–j). Of the antagonist compounds tested, the antagonist peptide UFP-101 and the small-molecule compound J-113397 produced robust competitive antagonism, fully reversing the activation of NOPLight at the concentrations used (1 μM), while nocistatin elicited a smaller decrease of the signal induced by N/OFQ. Importantly, none of the antagonistic ligands elicited a noticeable fluorescent response when applied alone to sensor-expressing cells. On the other hand, we could clearly detect positive fluorescent responses of NOPLight to several types of selective NOPR agonist compounds. In particular, the full agonist Ro 64 elicited the largest fluorescent response in this assay ($\Delta F/F_0 = 323\%$) and produced a response of similarly large magnitude in NOPLight-expressing primary neuronal cultures ($\Delta F/F_0 = 221\%$) (Supplementary Fig. 2c–f). Interestingly, all of the agonist compounds tested induced an overall smaller fluorescence response than N/OFQ itself, when applied at the same concentration (Fig. 1h).

We then characterized the spectral properties of NOPLight and NOPLight-ctr. Under both one-photon and two-photon illumination, NOPLight exhibited similar spectral characteristics as other GPCR-based sensors[41,45]. The sensor had an isosbestic point at around 440 nm, and peak performance, measured as the ratio between N/OFQ-bound versus unbound state, at 472 nm and 920/990 nm, respectively (Supplementary Fig. 3a, b). When tested with another NOPR agonist, RO 64-6198, the sensor exhibited similar one-photon spectral properties, whereas the control sensor showed little difference in excitation and emission in the presence or absence of the ligands (Supplementary Fig. 3a).

To evaluate the effect of p.H. change on the sensor, we measured the fluorescence intensity and response of NOPLight-expressing cells to N/OFQ when the cells were exposed to buffer solutions at set pH values (6–8). Under these conditions, there was no significant difference in sensor response across the tested pH range in comparison to neutral pH ($P = 0.942, 0.358, 0.883,$ and 0.289 for pH 6, 6.5, 7.5, 8, versus pH 7, respectively; one-way ANOVA with Tukey Kramer post hoc test, Supplementary Fig. 4a, b). We also acquired one-photon excitation and emission spectra of NOPLight under similar conditions. Within the tested pH range, the isosbestic point of NOPLight consistently fell within the range of 405–435 nm. Furthermore, the ratio between N/OFQ-bound versus unbound states also remained overall unaffected by the change in extracellular pH (Supplementary Fig. 4c). Lastly, we evaluated whether changes in neural activity could cause alterations in the observed fluorescence of NOPLight. To do this we co-expressed NOPLight with the red genetically encoded calcium indicator JRCaMP1b[46] in primary

cultured neurons via viral transduction, followed by simultaneous multiplexed imaging of NOPLight fluorescence and neuronal calcium activity in the absence and presence of bath-applied glutamate at a concentration known to evoke neuronal activity (5 μM)[47]. Overall, we did not observe noticeable changes in the fluorescence of NOPLight during periods of evoked neuronal activity (Supplementary Fig. 5), suggesting that potential alterations in the intracellular environment occurring during neuronal activity are not likely to influence the fluorescent responses of NOPLight.

The wild-type receptor of NOPLight, NOPR, is known to respond highly selectively to N/OFQ, as compared to all other endogenous opioid peptides[3,48]. To ensure that NOPLight retained a similar degree of ligand-selectivity, we tested its response to a series of opioid peptides applied to NOPLight-expressing cells at a high concentration (1 μM). NOPLight showed negligible response to dynorphins, enkephalins, and β-endorphin (Fig. 1i). Similarly, the sensor showed minimal response to a panel of fast neurotransmitters (Fig. 1j), indicating high N/OFQ ligand selectivity at NOPLight.

To ensure minimal interference of NOPLight with cellular physiology, we investigated the putative coupling of the sensor with downstream intracellular signaling pathways and compared it to that of wild-type human NOPR. Like other members of the opioid receptor family, NOPR is Gi/o coupled and inhibits basal and Gs-stimulated adenylate cyclase activity upon activation, thus lowering intracellular cAMP levels[3,48]. We used the GloSensor cAMP assay in HEK293 cells expressing either wild-type human NOPR or NOPLight to monitor intracellular cAMP level with a bioluminescence readout. Application of 1 nM N/OFQ significantly inhibited the forskolin-induced cAMP response in cells expressing the NOPR, while no effect was observed for NOPLight-expressing cells treated with up to 100 nM N/OFQ (Supplementary Fig. 6a). At higher concentrations of N/OFQ, inhibition of the cAMP signal in NOPLight-expressing cells was still significantly reduced compared to that of NOPR.

Under physiological conditions, activation of GPCRs can induce β-arrestin recruitment and/or receptor internalization. We monitored β-arrestin-2 recruitment to the cell surface upon N/OFQ stimulation using TIRF microscopy. Activation of NOPR induced strong β-arrestin-2 recruitment and subsequent internalization of the receptor, whereas NOPLight showed neither of these effects after prolonged occupancy by N/OFQ (Supplementary Fig. 6b–e). In accordance with the lack of coupling to β-arrestin-2, we also observed that the NOPLight response remained stable for over 1.5 h in the presence of N/OFQ, and the increase in fluorescence could be reversed by treating the cells with a membrane-impermeable peptide NOPR antagonist (UFP-101, 100 nM; Supplementary Fig. 6f, g). These results indicate that while NOPLight retains the ligand selectivity of the native receptor, its cellular expression has a very low likelihood of interfering with intracellular signaling, making the sensor suitable for utilization in physiological settings.

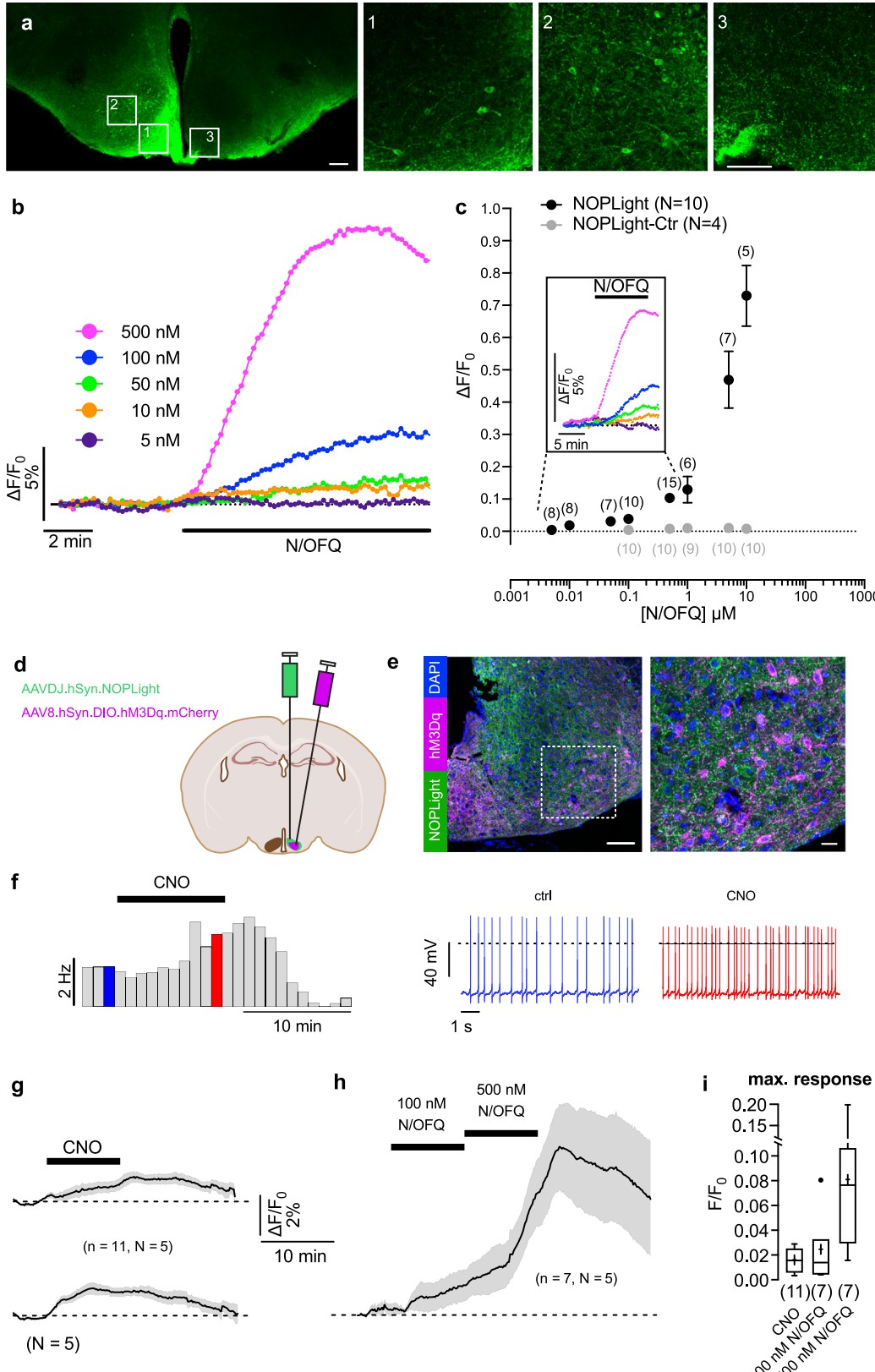

## Ex vivo characterization of NOPLight

We then expressed the sensor directly in brain tissue by injecting NOPLight-encoding adeno-associated virus (AAVDJ-hSyn-NOPLight) in the arcuate nucleus (ARC) of the hypothalamus of wild-type mice. After 4 weeks of incubation, the sensor was clearly expressed and was characterized for its functional response to exogenously perfused N/OFQ (Fig. 2a, b). NOPLight responses could be detected from ROIs in the ARC upon superfusion of as little as 10 nM N/OFQ on the slice ($\Delta F/F_0 = 1.9 \pm 1.1\%$, $n = 10$), and the magnitude of the responses continued to increase up to the highest N/OFQ concentration tested (10 µM, $\Delta F/F_0 = 73 \pm 21\%$) (Fig. 2c). To measure NOPLight OFF kinetics in brain tissue, we locally puffed N/OFQ next to the sensor-expressing area. The

**Fig. 2 | Ex vivo characterization of NOPLight. a** Expression of NOPLight in the ARC. Scale bar, 200 μm. Insets, scale bars, 50 μm. **b–f** Change in NOPLight fluorescence measured in ARC neurons in response to bath-applied N/OFQ or release of chemogenetically-activated PNOC neurons. Fluorescence was detected from 0.15–0.2 mm² ROIs. Each experiment was performed in a different brain slice. **b** NOPLight responses of a single ROI to increasing N/OFQ concentrations. **c** Concentration-response relation showing N/OFQ effect on the fluorescence of NOPLight (black) and NOPLight-Ctr (gray) -expressing cells in the ARC. Data represented as mean ± SEM. Inset, mean fluorescence increases in response to different concentrations of N/OFQ. Color-code as in (**b**). **d** Experimental schematic for chemogenetic experiments in brain slices. **e** Representative immunohistochemical image showing DREADD (hM3Dq) expression in PNOC neurons of the ARC as well as pan-neuronal expression of NOPLight. Scale bar, 100 μm. Magnification is shown on the right. Scale bar, 20 μm. **f** Perforated patch-clamp recordings from a hM3Dq-expressing PNOC neuron showing the effect of 10 min CNO

application (3 μM) on action potential frequency. Left, rate histogram (bin width 60 s). Right, representative sections of the original recording correspond to the times indicated by the blue and red color codes. **g–i** Changes (mean ± SEM) in NOPLight fluorescence measured in the ARC in response to activation of hM3Dq-expressing PNOC neurons by 10 min bath application of 3 μM clozapine-N-oxide (CNO) (**g**) and 100 nM and 500 nM N/OFQ (**h**). g Upper and lower are the same data: traces in the upper panel are aligned to the CNO application, and traces in the lower panel are aligned to the response onset. Recordings in (**g, h**) were performed from the same brain slices. Bars indicate the application of CNO and N/OFQ, respectively. Scale bars apply to (**g, h**). **i** Maximal fluorescence changes upon applications of CNO and N/OFQ, respectively. The box represents the interquartile range, indicating the 25th and 75th percentiles with a median marked by a horizontal line inside the box. Whiskers extend to the minimum and maximum values, excluding outliers (•). The numbers in brackets represent the number of experiments (brain slices). *N* values indicate the number of animals. Data represented as mean ± SEM.

sensor off kinetics ranged between 30–60 s under these conditions (30.9 ± 4.5 s, 41.1 ± 9.9 s, and 53.1 ± 6.6 s at 10, 50, and 500 nM, respectively, mean ± SEM) (Supplementary Fig. 7a–d). We next tested whether the in situ sensitivity of NOPLight would be sufficient to detect endogenous N/OFQ release in this ex vivo setting. For this purpose, we again used the preparation of ARC neurons with NOPLight expression and additionally expressed the activating DREADD hM3D in PNOC neurons (Fig. 2d, e). For this, a hM3Dq-encoding adeno-associated virus (AAV8-hSyn-DIO-hM3Dq-mcherry) was injected into the ARC of PNOC-Cre mice, enabling Cre-dependent expression of hM3Dq. hM3Dq activation by bath-applied clozapine-N-oxide (CNO) evoked a clear increase in the firing rate of PNOC neurons that lasted for several minutes (Fig. 2f). Correspondingly, we could detect an increase in NOPLight responses with a mean fluorescence change of ΔF/$F_0$ = 1.6 ± 0.3% (Fig. 2g–i), indicating that the sensor could report endogenous N/OFQ release under these conditions. The response of NOPLight was reversible and reflected the time course of chemogenetic activation of PNOC neurons. Overall, these results indicate that NOPLight provides a sensitive and specific readout of both superfused as well as endogenous N/OFQ peptide release in brain tissue.

### NOPLight activation by an exogenous NOPR agonist in vivo

Our results in vitro (Fig. 1g) showed potent and efficacious NOPLight responses to the small-molecule NOPR agonist Ro 64. Thus, we determined whether we could use fiber photometry to record NOPLight fluorescence in vivo and track the target engagement of this NOPR agonist action in real-time within the brain (Fig. 3). To do so, we injected WT mice with AAV-DJ-hSyn-NOPLight either in the VTA or in the ARC and implanted optic fibers above the injection sites for photometry recordings (Fig. 3a, b and Supplementary Fig. 8). At 3–4 weeks post-viral injection, we detected robust, dose-dependent increases in NOPLight fluorescence in both brain areas following systemic (i.p.) injection of increasing doses of Ro 64 (Fig. 3c and Supplementary Fig. 8a–d). To determine if the observed increase in NOPLight fluorescence in the VTA was produced by the sensor binding to the NOPR agonist, we pre-treated animals with two different selective NOPR antagonists, LY2940094 (LY, 10 mg/kg o.g.) or J-113397 (J11, 10 mg/kg i.p.) 30 min prior to injection of Ro 64. Both NOPR-selective antagonists fully inhibited the agonist-induced fluorescent signal (Fig. 3d, e).

Next, we injected additional cohorts of OPRL1-Cre (line characterized in the VTA in Supplementary Fig. 9a–h) or WT mice in the VTA with AAVs containing either a Cre-dependent variant of NOPLight (AAV-DJ-hSyn-FLEX-NOPLight), or the control sensor (AAV-DJ-hSyn-NOPLight-ctr) respectively, with optic fibers implanted above the injection site (Fig. 3f). We found that systemic injection of 10 mg/kg Ro 64 produced a robust increase in FLEX-NOPLight signal that was not significantly different from the agonist-induced signal we recorded from non-conditionally expressed NOPLight (Fig. 3g, h). In contrast, the control sensor showed no fluorescent response to a 10 mg/kg Ro

64 injection (Fig. 3g, h). These results indicate that NOPLight expressed in freely moving animals reliably provides dose-dependent and antagonist-sensitive detection of exogenous NOPR agonists in real-time.

### NOPLight detection of chemogenetically evoked endogenous N/OFQ release in the VTA

A primary goal motivating our development of the NOPLight sensor was to ultimately achieve real-time detection of local N/OFQ release in behaving animals. Thus, we set out to determine whether NOPLight reliably detects the endogenous release of N/OFQ. To accomplish this, we evoked endogenous N/OFQ release in a local paranigral VTA (pnVTA) circuit we previously identified[22] by using a chemogenetic approach to selectively activate VTA[PNOC] neurons while simultaneously recording changes in VTA-NOPLight fluorescence via fiber photometry. PNOC-Cre mice were co-injected in the VTA with two Cre-dependent AAVs containing i) AAV-DH-hSyn-NOPLight and ii) an mScarlet-tagged stimulatory hM3Dq DREADD (AAV5-EF1a-DIO-HA-hM3D(Gq)-mScarlet), with optic fibers implanted above the injection site (Fig. 4a). Control animals received an mCherry (AAV5-EF1a-DIO-mCherry) injection in place of the red fluorophore-tagged DREADD. Based on our earlier spectral characterization which showed that the isosbestic point of NOPLight is closer to 440 nm (Supplementary Fig. 3), we tested and characterized an alternative set of LED wavelengths in this group of experiments, using 435 and 490 nm as the isosbestic and signal wavelengths, respectively. Activation of the hM3Dq DREADD via systemic injection with 5 mg/kg of clozapine-N-oxide (CNO) produced a significant increase in NOPLight fluorescence that was not observed in the control animals expressing mCherry in place of the DREADD (Fig. 4b). To confirm that this increase was truly the result of chemogenetically evoked endogenous N/OFQ release and thus acting via a NOPR-dependent mechanism, pretreatment with the selective NOPR antagonist LY2940094[49] (LY, 10 mg/kg, o.g.) 30 min prior to CNO injection prevented the CNO-induced increase in NOPLight fluorescence (Fig. 4c). Together, these results provide strong evidence that NOPLight can detect evoked endogenous release of N/OFQ in freely moving animals.

### NOPLight detects dynamics in endogenous N/OFQ VTA tone during head-fixed consummatory behaviors in vivo

We also examined NOPLight's ability to report transient, endogenous N/OFQ release evoked by different naturalistic behavioral states. N/OFQ and its receptor NOPR have been implicated in the neuromodulation of a wide variety of essential behavioral processes including stress, aversion, motivation, reward seeking, and feeding[22,23,27,50–55]. We have previously identified a role for VTA N/OFQ signaling in reward-related and aversive behavior, and using GCaMP found that pnVTA[PNOC] neurons activity is suppressed during sucrose consumption[22]. Therefore, we injected WT mice with AAV-DJ-hSyn-NOPLight in the VTA,

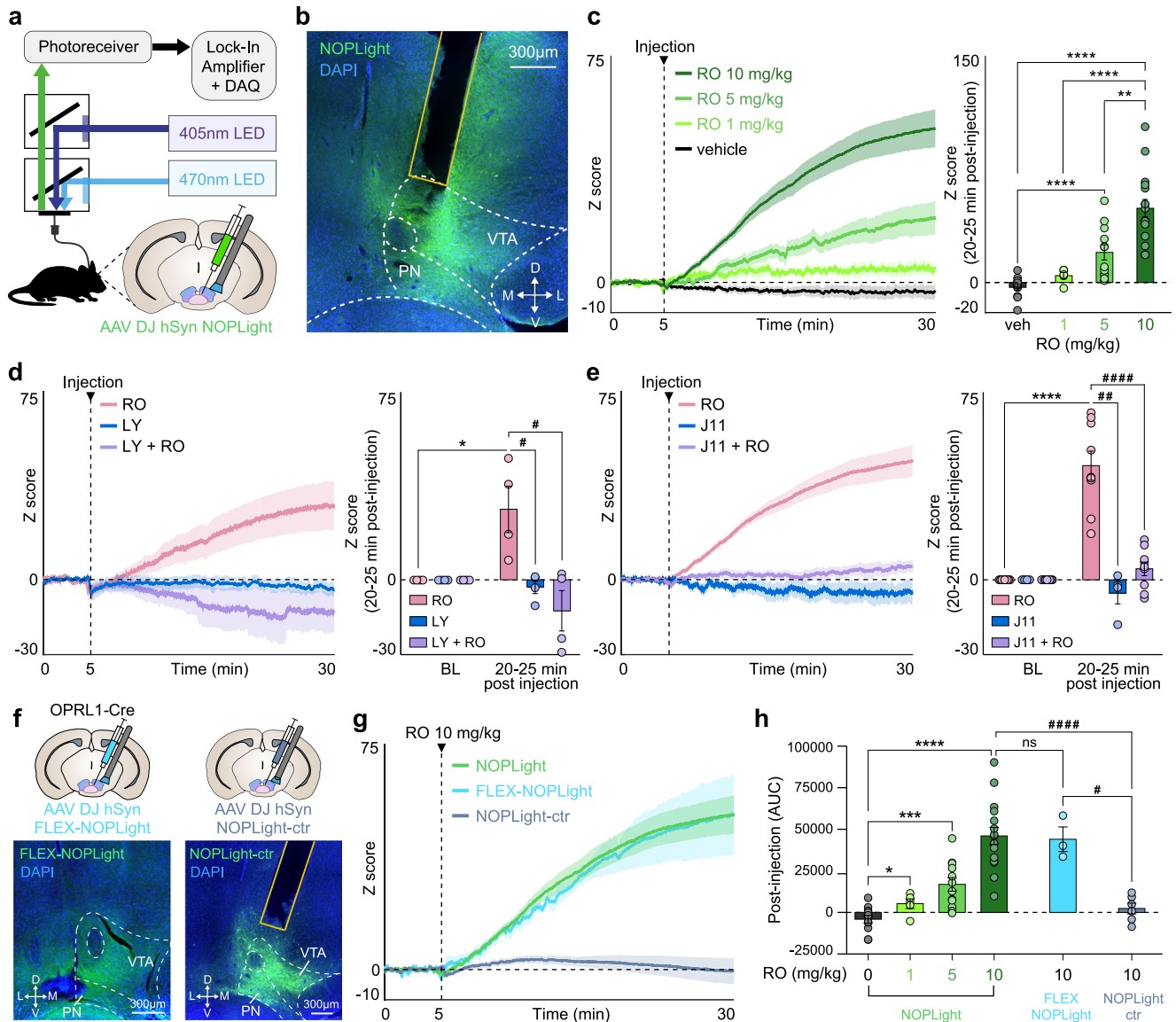

**Fig. 3 | Characterization of NOPLight response in vivo to pharmacological agonism and antagonism. a** Fiber photometry schematic. Cartoon of NOPLight viral injection and fiber implant in the VTA. **b** Representative image showing DAPI (blue) and NOPLight (green) expression with fiber placement. **c** Left: Averaged traces of NOPLight fluorescence after systemic (i.p.) injection of vehicle (veh, black; $n = 11$ mice) or 1, 5, or 10 mg/kg of selective NOPR agonist Ro 64-6198 (RO, green; $n = 5$, 12, or 16 mice/group, respectively). Right: Mean NOPLight fluorescence 20–25 min after RO injection increases dose-dependently (two-tailed Mann–Whitney test, **$p = 0.0012$, ****$p < 0.0001$). Data represented as mean ± SEM. **d** Left: Averaged traces of NOPLight fluorescence after 10 mg/kg RO (pink), 10 mg/kg selective NOPR antagonist LY2940094 (LY, o.g., blue), or both (purple) ($n = 4$ mice). Right: Increase in signal 20–25 min after RO is blocked by LY pretreatment (two-tailed Mann–Whitney test, *$p = 0.0286$, #$p = 0.0286$, $n = 4$ mice). Data

represented as mean ± SEM. **e** Same as (**d**) for NOPR antagonist J-113397 (J11, i.p.) (two-tailed Mann–Whitney test, ##$p = 0.0028$, ####$p < 0.0001$, ****$p < 0.0001$, $n = 9$ mice, RO; 4 mice, J11; 9 mice, J11 + RO). Data represented as mean ± SEM. **f** Top: Cartoon of FLEX-NOPLight or NOPLight-ctr viral injection and fiber implant in the VTA. Bottom: Representative image showing expression of DAPI (blue) and FLEX-NOPLight (left, green) or NOPLight-ctr (right, green), with fiber placement in VTA. **g** Averaged traces of NOPLight (green), FLEX-NOPLight (blue), or NOPLight-ctr (gray) fluorescence ($n = 16$, 3, 6 mice, respectively) after systemic (i.p.) injection of 10 mg/kg RO. Data represented as mean ± SEM. **h** Area under the curve (AUC) of each NOPLight variant after RO injection (two-tailed Mann–Whitney test, *$p = 0.0275$, ***$p = 0.0001$, ****$p < 0.0001$, ns not significant ($p = 0.9577$), #$p = 0.0238$, ####$p < 0.0001$. Group sizes left to right: $n = 11$, 5, 12, 16, 3, and 6 mice). Data represented as mean ± SEM. PN paranigral VTA, veh vehicle, BL baseline.

implanted optical fibers above the injection site, and secured stainless-steel rings to allow for head-fixed fiber photometry recording of NOPLight during sucrose consumption (Fig. 5a). Mice were food-restricted (85–90% of their starting body weight) and then NOPLight signal was recorded during cued access to 10% sucrose solution (15 trials, 5 s sucrose access/trial) (Fig. 5b). Sucrose trials were cued by a 5-second auditory tone (4 kHz, 80 dB) that preceded the 5 s of sucrose access to determine if NOPLight would respond to any salient stimulus rather than specifically to sucrose consumption. We found that NOP-Light fluorescence remained unchanged during the 5-second tone, but

significantly decreased during sucrose consumption (Fig. 5c–f). In contrast, animals that were pre-treated with the selective NOPR antagonist J-113397 (J11, 20 mg/kg, i.p.) 30 min prior to cued-sucrose recordings had no change in NOPLight signal during sucrose consumption (Fig. 5c, f). Mice made a similar number of licks on the sucrose spout during J-113397 and vehicle sessions, suggesting that the lack of signal change during sucrose consumption following J-113397 treatment was a result of the NOPR antagonist blocking NOPLight's detection of endogenous N/OFQ levels. This result is consistent with our previous study, which showed that nociceptin-containing neurons

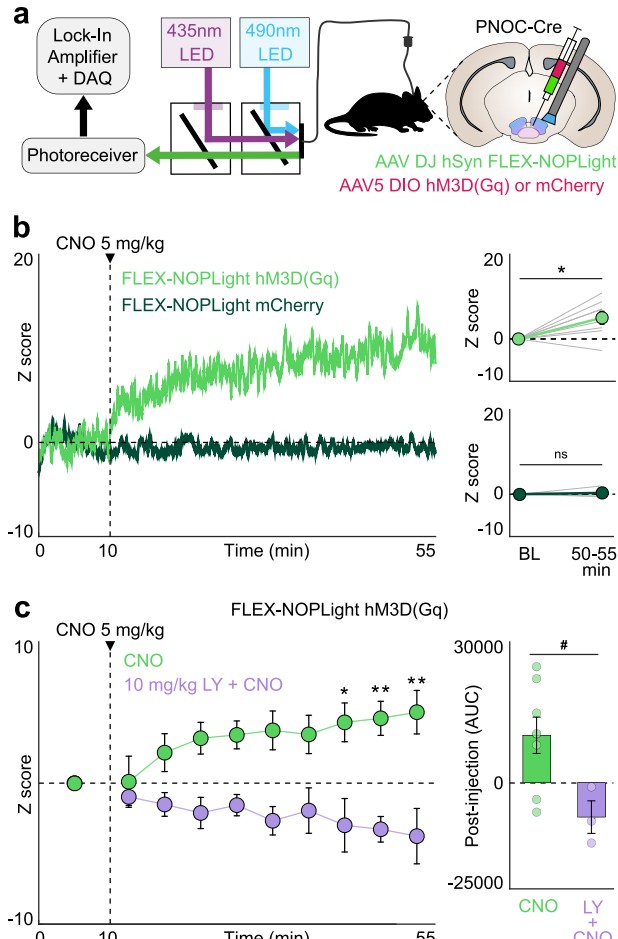

**Fig. 4 | NOPLight detects chemogenetically evoked endogenous N/OFQ release in vivo. a** Schematic of fiber photometry setup. Coronal brain cartoon of fiber implant and viral co-injection of FLEX-NOPLight with either DIO-hM3D(Gq) or mCherry in the VTA of PNOC-Cre mice. **b** Left: Representative traces of FLEX-NOPLight fluorescence after systemic (i.p.) injection of 5 mg/kg clozapine-N-oxide (CNO) in hM3D(Gq) (light green) or mCherry control (dark green) animals. Right: Mean FLEX-NOPLight fluorescence 40–45 min after CNO injection is significantly elevated relative to the pre-injection baseline period (BL) in hM3D(Gq) (two-tailed Wilcoxon test, $*p = 0.0391$, $n = 8$ mice) but not control animals (two-tailed Wilcoxon test, $p > 0.9999$, ns not significant, $n = 3$ mice). Z-scores for each individual animal averaged at baseline, and 50–55 min are shown (gray lines). Data represented as mean ± SEM. **c** Left: FLEX-NOPLight fluorescence (green) averaged before injection of 5 mg/kg CNO (0–10 min), and in 5 min bins following injection. 10 mg/kg of selective NOPR antagonist LY2940094 (LY) was administered (o.g.) to the LY + CNO (purple) group 30 min prior to photometry recording (two-way repeated-measures ANOVA with Bonferroni's post hoc test, $*p = 0.0207$ (40–45 min), $**p = 0.0094$ (45–50 min) or 0.0024 (50–55 min), $n = 8$ mice, CNO; three mice, LY + CNO). Right: Area under the curve (AUC) of FLEX-NOPLight signal after CNO injection (cumulative, 45 min). Pretreatment with NOPR antagonist LY prevents CNO-induced increases in FLEX-NOPLight fluorescence (two-tailed Mann–Whitney test, $^\#p = 0.0485$, $n = 8$ mice, CNO; three mice, LY + CNO). Data represented as mean ± SEM.

have lower activity during reward consumption. These results demonstrate that NOPLight is sensitive to changes in the endogenous N/OFQ tone in vivo within the VTA, in behaviorally relevant appetitive contexts.

## NOPLight detects natural endogenous N/OFQ release following aversive stimuli in vivo

In addition to having a well-established role in reward processing, the VTA is known to mediate aversive states. Given that N/OFQ and NOPR

are also widely implicated in stress and aversive responses, we evaluated whether NOPLight could detect endogenous N/OFQ release in the VTA during an aversive behavior in freely moving animals. Our previous findings showed that stimulating pnVTA[PNOC] neurons drives aversive responses, so we predicted that the VTA N/OFQ system would be engaged by an aversive stimulus. To test this, we used fiber photometry to record pnVTA[PNOC] activity in response to tail suspension (Fig. 5g). PNOC-Cre mice injected with AAVDJ-EF1a-DIO-GCaMP6m and implanted with optic fibers in the VTA underwent four trials of a 10-s tail suspension, resulting in a robust increase in GCaMP6m fluorescence lasting for the duration of the suspension (Fig. 5h, i). To determine if the increase in pnVTA[PNOC] calcium activity was reflected by NOPLight as an increase in N/OFQ release in the VTA, we repeated the tail suspension test in OPRL1-Cre mice injected with AAV-DJ-hSyn-FLEX-NOPLight and WT mice injected with AAV-DJ-hSyn-NOPLight-ctr (Fig. 5g). Fiber photometry recordings of FLEX-NOPLight showed an increase in fluorescence during tail suspension that was significantly elevated in comparison to NOPLight-ctr (Fig. 5j, k). These data indicate that NOPLight can reliably report the endogenous release of N/OFQ during aversive behavioral events within the VTA.

## Monitoring endogenous N/OFQ dynamics in reward-seeking operant behavior

Our prior work extensively characterized the calcium activity of pnVTA[PNOC] neurons during operant responding for reward and identified a role for VTA N/OFQ signaling in constraining motivation to obtain a reward. Therefore, we sought to determine whether our sensor could report the dynamics of endogenous N/OFQ release in the VTA during operant reward-seeking behavior. WT mice injected with AAV-DJ-hSyn-NOPLight and implanted with optic fibers in the VTA were food-restricted (~85–90% of their starting body weight) and then trained on a Pavlovian and operant conditioning schedule (Fig. 6a). Mice were first trained in Pavlovian conditioning sessions to associate a 5-second house light cue (CS) with the delivery of a sucrose pellet (US) (Supplementary Fig. 10a). Next, they were trained on a fixed ratio (FR) operant schedule, learning first to perform one (FR1) and later three (FR3) nose pokes into an active port to trigger the light cue and sucrose reward. Mice successfully learned that only nose pokes in the active port would result in reward delivery (Supplementary Fig. 10b). Tracking of pellet consumption across the training paradigm also confirmed that mice consumed the majority of rewards they obtained across sessions (Supplementary Fig. 10c).

NOPLight fluorescent signals recorded during early Pavlovian conditioning sharply increased in response to the onset of the light cue (Supplementary Fig. 10d), which is consistent with fiber photometry recordings of the calcium activity (GCaMP6s) of a posterior population of pnVTA[PNOC] neurons known to project locally within the VTA[22]. Across all Pavlovian and operant conditioning schedules, we observed a robust decrease in NOPLight fluorescence that persisted throughout the reward consumption period, and was immediately followed by a transient increase in signal upon the end of a feeding bout (Supplementary Fig. 10d–h). This decrease in signal during reward consumption and subsequent increase at the end of the feeding period is consistent with previously reported calcium activity patterns of posterior pnVTA[PNOC] neurons recorded during both Pavlovian and operant conditioning. Importantly, during FR3 recordings when mice performed an active nose poke that did not yet meet the threshold for a reward delivery, we observed an increase in NOPLight fluorescence following the nose poke instead of a decrease, indicating that the decrease is related to reward consumption and not the operant action (Supplementary Fig. 10i). Taken together, NOPLight signal in the VTA closely resembles known patterns of pnVTA[PNOC] neuron GCaMP activity during Pavlovian and operant reward-related behaviors.

After completing FR3 operant training, mice were placed in a progressive ratio (PR) test where the required nose poke criterion

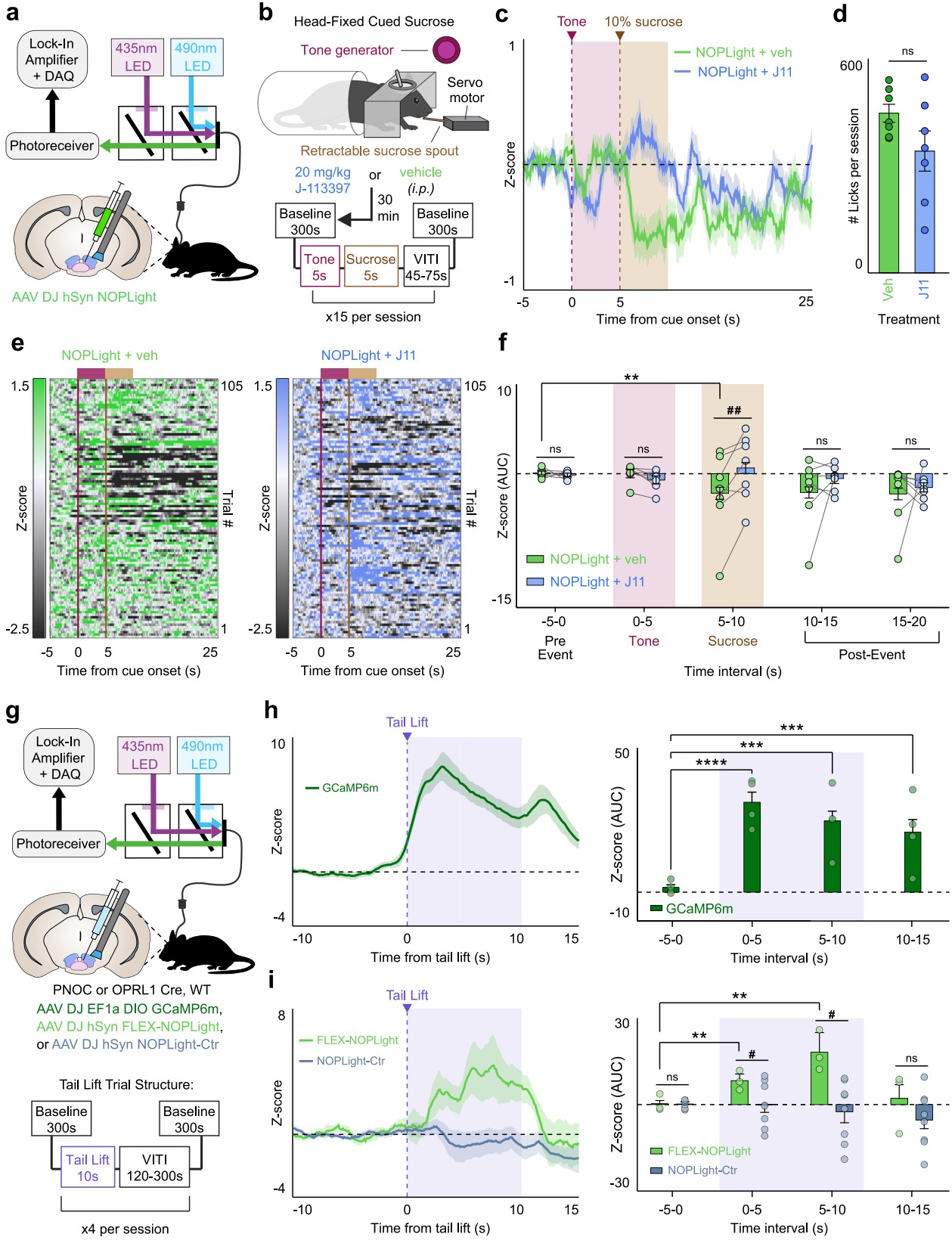

increases exponentially with each subsequent reward until the animal reaches a motivational breakpoint where they are unwilling to exert any further effort to obtain a reward (Fig. 6b). We extracted fiber photometry epochs surrounding active nose pokes that were reinforced with delivery of a sucrose pellet reward, finding that NOPLight fluorescence decreases during reward consumption in the PR test and

transiently increases immediately upon completion of a consumption bout (Fig. 6c, f). In contrast, NOPLight signal remained stable during epochs where animals performed an unsuccessful active nose poke that did not result in reward delivery (Fig. 6d, g). NOPLight signal during reward consumption is significantly suppressed in comparison to time periods where animals expected but did not obtain a reward

**Fig. 5 | NOPLight in vivo reports bidirectional endogenous N/OFQ dynamics during consummatory and aversive behaviors. a** Fiber photometry schematic. Cartoon of viral injection of NOPLight and fiber implant in VTA of wild-type mice (*n* = 7 mice). **b** Cartoon depicting head-fixed cued-sucrose setup and trial structure. **c** Averaged traces of NOPLight fluorescence following pretreatment with vehicle (veh, green) or 20 mg/kg NOPR antagonist J-113397 (J11, blue), aligned to tone onset (magenta, shaded). Data represented as mean ± SEM. **d** Average licks made during vehicle or J-113397 sessions (two-tailed Wilcoxon test, *p* = 0.1649, ns not significant, *n* = 7 mice). Data represented as mean ± SEM. **e** Heat maps of NOPLight fluorescence, rows correspond to trials averaged in (**c**) for vehicle (left) and J11 (right) sessions. **f** Area under the curve (AUC) for averaged traces from (**c**), calculated over 5-s intervals surrounding cued-sucrose events. Decrease in NOPLight fluorescence during 10% sucrose access (two-tailed Wilcoxon test, **p* = 0.0034, *n* = 7 mice) is blocked by J11 pretreatment (two-tailed Mann–Whitney test, ##*p* = 0.0022, *n* = 7

mice). Data represented as mean ± SEM. **g** Top: Fiber photometry schematic. Middle: Cartoon depicting fiber implant and viral injection of DIO-GCaMP6m, FLEX-NOPLight, or NOPLight-ctr into the VTA of PNOC-Cre, OPRL1-Cre, or WT mice, respectively. Bottom: Session trial structure. **h** Left: Averaged trace of pnVTA^PNOC GCaMP6m activity during tail lift. Right: AUC for photometry trace calculated over 5-s intervals surrounding tail lift (two-tailed Wilcoxon test, ***p* = 0.0002 (5–10 s) or 0.0003 (10–15 s); ****p* < 0.0001, *n* = 4 mice). Data represented as mean ± SEM. **i** Left: Averaged traces of FLEX-NOPLight (green) and NOPLight-ctr (gray) fluorescence during the tail lift. Right: AUC for photometry traces calculated over 5-s intervals surrounding tail lift. FLEX-NOPLight fluorescence increases during tail lift (two-tailed Wilcoxon test, ***p* = 0.0034 (0–5 s) or 0.0093 (5–10 s); *n* = 3 mice), NOPLight-ctr fluorescence remains unchanged (two-tailed Mann–Whitney test, #*p* = 0.0264 (0–5 s) or 0.0326 (5–10 s); *n* = 8 mice, NOPLight-ctr; 3 mice, FLEX-NOPLight). Data represented as mean ± SEM. VITI variable inter-trial interval.

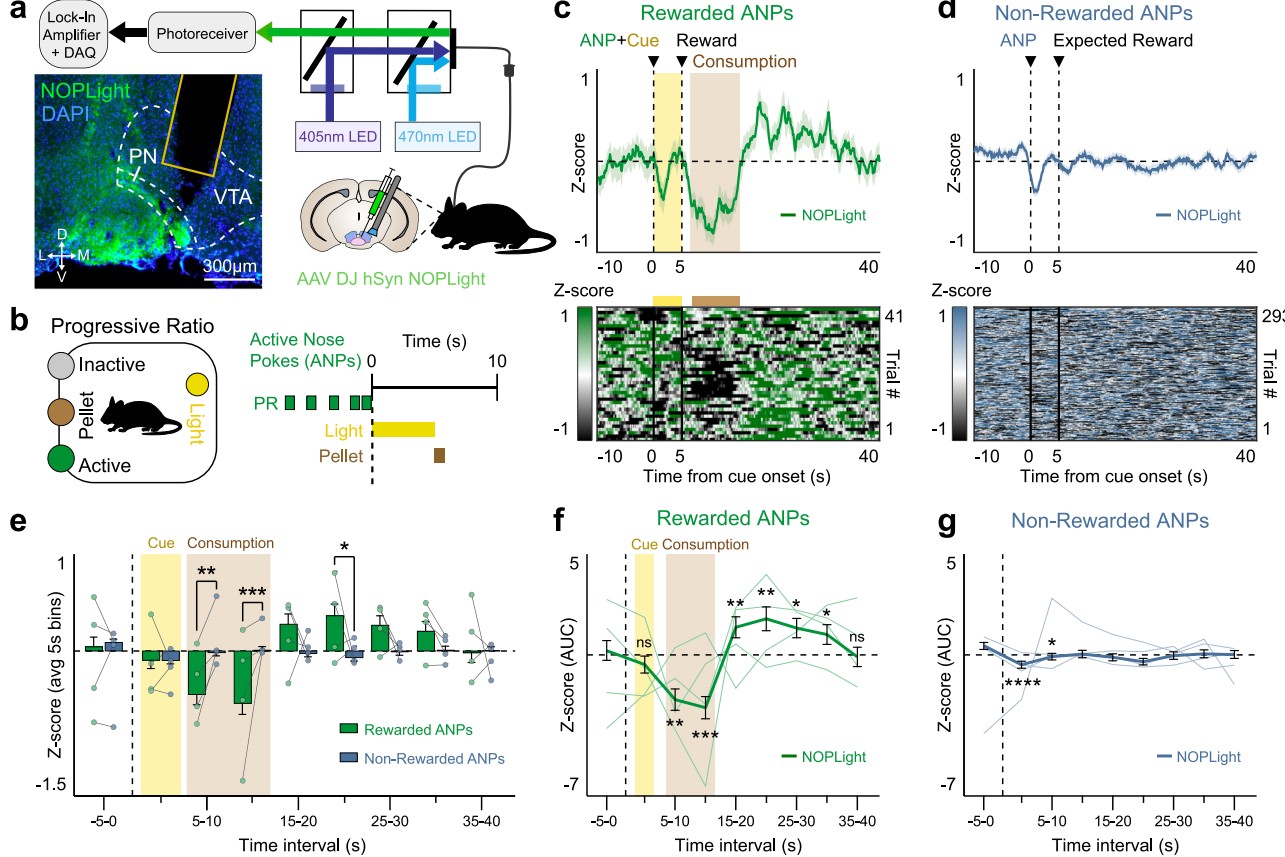

**Fig. 6 | NOPLight detection of endogenous VTA N/OFQ release during high effort reward-seeking. a**. Fiber photometry schematic. Cartoon of NOPLight viral injection and fiber implant in the VTA of WT mice (*n* = 4 mice). Representative image showing expression of DAPI (blue) and NOPLight (green) with fiber placement in VTA. **b**. Left: Cartoon depicting operant box setup for the progressive ratio (PR) test. Right: Trial structure for reward delivery during the paradigm. **c**. Top: Averaged trace of NOPLight signal for all mice (*n* = 4), aligned to rewarded active nose pokes (ANPs) made during the PR test. Epoch includes the 5 s light cue (yellow, shaded) that precedes reward delivery. Time to pellet retrieval and duration of consumption period averaged across all trials (brown, shaded). Data represented as mean ± SEM. Bottom: Heat map, each row corresponds to a single, rewarded ANP epoch. **d** Top: Averaged trace of NOPLight signal for all mice (*n* = 4), aligned to non-rewarded ANPs made during the PR test. Data represented as mean ± SEM. Bottom: Heat map, each row corresponds to a single, non-rewarded ANP epoch. **e** NOPLight

fluorescence averaged over 5-s intervals for rewarded (green) and non-rewarded (blue) ANP epochs. Time to pellet retrieval and duration of consumption period averaged across all rewarded trials (brown, shaded). Across rewarded ANP trials, NOPLight signal decreases during reward consumption then immediately increases post-consumption relative to NOPLight signal measured during non-rewarded ANP epochs (two-way repeated-measures ANOVA with Bonferroni's post hoc test, **p* = 0.0122, ***p* = 0.0033, ****p* = 0.0003, *n* = 4 mice). Data represented as mean ± SEM. **f, g** Area under the curve (AUC) for photometry traces from **c** and **d**, respectively, calculated over 5-s intervals surrounding (**f**) rewarded ANPs (two-tailed Wilcoxon test, **p* = 0.0125 (25–30 s) or 0.031 (30–35 s); ***p* = 0.007 (5–10 s) or 0.0042 (15–20 s); ****p* = 0.0008 (10–15 s); *n* = 4 mice) and (**g**) non-rewarded ANPs (two-tailed Wilcoxon test, **p* = 0.0128 (5–10 s), ****p* < 0.0001 (0–5 s), *n* = 4 mice). Data represented as mean ± SEM.

(Fig. 6e). Notably, our detection of negative changes to NOPLight fluorescence relative to baseline suggests the presence of N/OFQ tone in the VTA, which is consistent with our previous findings regarding this circuit. Collectively, these results indicate that NOPLight faithfully

reports bidirectional changes in N/OFQ release in the VTA during freely moving behaviors. Our findings demonstrate NOPLight's utility in detecting endogenous N/OFQ release during discrete behavioral epochs within the VTA and potentially other N/OFQ brain regions.

## Discussion

Here we describe the engineering, characterization, and application of a genetically encoded sensor (NOPLight) for monitoring the opioid neuropeptide N/OFQ in vitro, ex vivo, and in vivo. Endogenous opioid peptides represent one of the largest classes of neuropeptide families, yet detecting their release, dynamics, and properties in vitro and in vivo has been a challenge for over 60 years since their discovery[56–58]. We sought to develop a sensor which could detect (1) evoked release, (2) endogenous release during naturalistic behavior, and (3) exogenous ligands in vivo to inform brain localization of pharmacological agents. These properties have been long sought after to better understand neuropeptide transmission generally, and more specifically opioid peptides and their actions.

The neuropeptide biosensor we developed exhibits a large dynamic range both in HEK293T cells and in neurons, subsecond activation kinetics, very high ligand selectivity, a similar pharmacological profile to that of NOPR, and no detectable interference with endogenous signaling pathways. We demonstrated that NOPLight dose-dependently responds to systemic administration of a NOPR agonist, with sensitivity to blockade by selective NOPR antagonists. NOPLight is also capable of detecting an endogenous release of N/OFQ evoked by either chemogenetic stimulation (hM3Dq DREADD) of PNOC neurons or during natural behavior.

N/OFQ and NOPR are widely implicated in diverse behavioral states, which was reflected by our fiber photometry recordings of NOPLight during varied behaviors. Our recordings of the NOPLight signal during head-fixed behavior revealed a NOPR antagonist-sensitive decrease in N/OFQ signaling time-locked with access to a 10% sucrose solution. We also identified an increase in N/OFQ release in response to an aversive tail suspension using NOPLight, which was not present in recordings of our control variant of the sensor. While it has been suggested that N/OFQ is released in response to aversive or stressful stimuli, our findings are the first to our knowledge to directly detect endogenous N/OFQ release during an aversive response.

Previous GCaMP recordings of PNOC neurons with local input in the VTA during operant conditioning tasks revealed dynamic engagement of pnVTA^PNOC neurons during reward-seeking and consumption behavior[22]. While in many cases, calcium mobilization is required for dense-core vesicle fusion[59], calcium activity is not a direct correlate for peptide release, and as such, the dynamics of released N/OFQ could not be established in this prior study. Here we report NOPLight activity in the VTA in reward-seeking behavior during fixed ratio -1, −3, and progressive ratio paradigms, which identified a rapid and sustained decrease during reward consumption, and a transient increase after consumption had ended (Fig. 6 and Supplementary Fig. 10). This pattern of NOPLight signal closely resembles the expected dynamics of N/OFQ release based on the prior study's GCaMP recordings[22]. Notably, we observed both dynamic increases and decreases in NOPLight fluorescence during behavioral epochs, suggesting that the NOPLight can be used to detect changes in peptide tone over behaviorally relevant timescales. These data also provide the ability to align neuronal activity measured either by calcium dynamics or electrophysiology with neuropeptide release during freely moving behavioral epochs. Given the recent discoveries that PNOC and NOPR are important for motivation, feeding, and sleep induction[22,23,50–53], understanding the dynamic properties of this opioid peptide system is now of even greater importance as this receptor is now considered a major target for insomnia, addiction, and depression[49,60,61].

The endogenous activity of PNOC neurons in the VTA is thought to provide inhibitory tone onto VTA dopamine neurons, constraining motivation to seek rewards[15,22,24,62–64]. Consistent with this concept, here we observed a decrease in NOPLight fluorescence when animals engaged in reward consumption. Additionally, we measured an increase in NOPLight signal immediately upon completion of a reward consumption bout, suggesting that N/OFQ is released in the VTA after animals have consumed an obtained reward. Since NOPR is largely expressed on dopamine neurons in the VTA and exerts inhibitory influence over their activity[24,62], it is possible that this increase in endogenous N/OFQ release after consumption reflects a temporarily satiated state where N/OFQ signaling transiently increases to suppress tonic dopamine neuron activity, thus reducing motivation to seek out additional rewards. These findings provide insight into the dynamics of N/OFQ signaling in the VTA which acts to coordinate motivated behavior through dopaminergic interactions. Future studies will be able to employ NOPLight in concert with the recently developed red-shifted dopamine sensors[65] and other N/OFQ-selective tools like the new OPRL1-Cre line (Supplementary Fig. 9) to improve our understanding of N/OFQ modulation of dopamine circuitry during reward-seeking.

We tested the performance of the sensor in vivo using different wavelength pairs. Following the development of GCaMPs, sensor excitation has conventionally employed a 405 nm wavelength light as a read-out for non-ligand sensitive signals (known as the isosbestic channel), that typically acts as an internal control for fiber photometry experiments, particularly in freely moving animals[66,67]. As a result, most commercially available photometry setups are tailored to accommodate this wavelength. In this work, we noted that, based on the results from our spectral characterization of NOPLight, excitation at a wavelength of 435 nm is better suited as a control 'isosbestic' channel. It is particularly worth noting that many recently developed intensiometric GPCR-based biosensors exhibit a similar spectral property with a right-shifted isosbestic point (i.e., >420 nm)[45,68–71]. In these cases, the use of 405 nm as the isosbestic channel for these sensors may lead to confounding results or difficulty in interpretation in some contexts, such as regions where N/OFQ release in response to a given behavior is relatively low, or during behaviors that are highly subject to motion artifact. Since it can be cost and time-prohibitive for research groups to add an alternative recording parameter to existing photometry setups, we demonstrated that we were still able to detect meaningful changes in NOPLight fluorescence when using the 405 nm wavelength in some of our fiber photometry recordings. Furthermore, head-fixation and red fluorophore-based motion controls are commonly used as alternatives to isosbestic controls that could easily be implemented with NOPLight[67,72]. Our careful evaluation of NOPLight's performance, when recorded with different isosbestic and excitation wavelengths, provides valuable insight into its photophysical properties that will help inform the successful application of NOPLight and other neuropeptide sensors in future studies. Our work here lays the critical groundwork that will need to be built upon through continued use of NOPLight and its control variant in similar reward-related behaviors, different brain regions, and with additional controls, which will be important for optimizing their implementation and developing general best practices amidst the rapidly expanding use of fluorescent peptide sensors.

Our findings present NOPLight as an approach to improve investigations of endogenous opioid peptide dynamics with high spatiotemporal resolution. We characterized NOPLight expression, selectivity, and sensitivity to endogenous N/OFQ release both in vivo and in vitro. Future optimization of the sensor should seek to improve quantum yield (fluorescent readout) at lower peptide concentrations and to develop red-shifted variants to provide more flexibility in multiplexing NOPLight with other optical tools and sensors. This sensor directly helps to address a longstanding limitation in understanding the real-time dynamics of endogenous peptide release during behavioral epochs. Future applications of neuropeptide sensors such as NOPLight will advance our understanding of the underlying mechanisms by which endogenous opioid peptides control, stabilize, and modulate neural circuits to regulate behavior.

## Methods

### Ethical statement

Animal procedures were performed in accordance with the guidelines of the European Community Council Directive or the Animal Welfare Ordinance (TSchV 455.1) of the Swiss Federal Food Safety and Veterinary Office and were approved by the Zürich Cantonal Veterinary Office, and the other respective local government authorities (Bezirksregierung Köln, Animal Care and Use Committee of University of Washington).

### Molecular cloning and structural modeling

The prototype sensor was designed in silico by sequence alignment (Clustal Omega2) and ordered as a geneblock (Thermo Fisher) flanked by *Hind*III and *Not*I restriction sites to be subsequently cloned into pCMV vector (Addgene #111053). For sensor optimization, site-directed mutagenesis and Circular Polymerase Extension Cloning was performed by polymerase chain reaction with custom-designed primers (Thermo Fisher) using a Pfu-Ultra II fusion High-Fidelity DNA Polymerase (Agilent). Sanger sequencing (Microsynth) was performed for all constructs reported in the manuscript. The structural prediction of the NOPLight was generated by a deep learning-based modeling method, RoseTTAFold[4].

### Cell culture, confocal imaging, and quantification

HEK293T cells (ATCC CRL-3216) were authenticated by the vendor. They were seeded in glass bottom 35-mm (MatTek, P35G-1.4-14-C) or 24-well plates (Cellvis, P24-0-N) and cultured in Dulbecco's modified Eagle's medium (Gibco) with 10% fetal bovine serum (Gibco) and antibiotic-antimycotic (1:100 from 10,000 units/ml penicillin; 10,000 μg/ml streptomycin, 25 μg/ml amphotericin B, Gibco) mix at 37 °C and 5% $CO_2$. Cells were transduced at 70% confluency using the Effectene transfection kit (QIAGEN) and imaged after 24–48 h. Primary neuronal cultures were prepared as follows: the cerebral cortex of 18 days old rat embryos was carefully dissected and washed with 5 ml sterile-filtered PBGA buffer (PBS containing 10 mM glucose, 1 mg/ml bovine serum albumin, and antibiotic-antimycotic 1:100 (10,000 units/ml penicillin; 10,000 μg/ml streptomycin; 25 μg/ml amphotericin B)). The cortices were cut into small pieces with a sterile scalpel and digested in 5 ml sterile-filtered papain solution for 15 min at 37 °C. The supernatant was removed, and tissue was washed twice with complete DMEM/FCS medium (Dulbecco's Modified Eagle's Medium containing 10% fetal calf serum and penicillin/streptomycin, 1:100). Fresh DMEM/FCS was then added, and the tissue gently triturated and subsequently filtered through a 40-μm cell-strainer. Finally, the neurons were plated at a concentration of 40,000–50,000 cells per well onto the poly-L-lysine (50 μg/ml in PBS) coated 24-well culture plate and incubated overnight at 37 °C and 5% $CO_2$. After 24 h of incubation, the DMEM medium was replaced with freshly prepared NU-medium (Minimum Essential Medium (MEM) with 15% NU serum, 2% B27 supplement, 15 mM HEPES, 0.45% glucose, 1 mM sodium pyruvate, 2 mM Gluta-MAX). Cultured neurons were transduced at 4 days in vitro (DIV4) with AAV-DJ-hSynapsin1-NOPLight, AAV1-hSyn1-NES-jRCaMP1b, or AAV-DJ-hSynapsin1-NOPLight-ctr viruses in culture media using a final titer for each virus between $4 \times 10^9$ and $4 \times 10^{10}$ GC/ml, and were imaged between DIV19–21. All reagents used are from Gibco. Unless otherwise noted, confocal imaging for all constructs reported in the manuscript are performed as follows:

Images were acquired on an inverted Zeiss LSM 800 microscope with a 488-nm laser for NOPLight and NOPLight-ctr, and a 564-nm laser for Red-Dextran dye and JRCaMP1b. For characterization of the dynamic range, expression level, and pharmacological properties, HEK293T cells and/or neurons expressing the construct were first rinsed with HBSS (Gibco) and imaged at a final volume of 100 μL HBSS under a 40x objective. For testing sensor performance in vitro in various pH, NOPLight-expressing HEK293T cells were first incubated in PBS (Gibco) with adjusted pH (6–8) for 3 min before the addition of the ligand. For pharmacological characterizations, the following compounds were used: Nocistatin (Abbiotec); J-113397 (Sigma-Aldrich); UFP-101 (Sigma-Aldrich); Ro 64-6198 (Sigma-Aldrich); MCOPPB (Cayman); Orphanin-FQ (1–11) (Tocris); Leu-Enkephalin (Cayman); Met-Enkephalin (Cayman); Dynorphin A (Cayman); Dynorphin-B (Cayman); β-Endorphin (Sigma-Aldrich); γ-Aminobutyric acid (Sigma-Aldrich); Dopamine hydrochloride (Sigma-Aldrich), Acetylcholine bromide (Sigma-Aldrich), Glutamate (Sigma-Aldrich). All compounds were diluted to the desired final concentration in HBSS before the experiment except Ro 64-6198 and J-113397, which were diluted in <0.02% DMSO. All ligands were carefully pipetted into the imaging buffer during the experiment. To determine the apparent affinity of the sensor, HEK293T cells and neurons cultured in glass bottom 24-well plates were rinsed with HBSS and imaged under a 20x objective with a final buffer volume of 500 μL HBSS. Ligands were manually applied to the cells during imaging to reach the desired final concentration.

ΔF/$F_0$ was determined as the ratio of change in fluorescence signal change upon ligand activation and the baseline fluorescence level

$$\frac{F_t - F_0}{F_0} \tag{1}$$

where $F_0$ is determined as the mean intensity value over the baseline imaging period.

$$F_0 = \frac{1}{n}\sum_{t=0}^{t=n} F_t \tag{2}$$

Unless stated otherwise, only pixels corresponding to cell membrane were considered as regions of interest (ROIs) thus included in the analysis. ROIs were selected by auto-thresholding function of ImageJ and confirmed by visual inspection.

### Kinetic measurements and analysis

To obtain the time constant for sensor activation, red fluorescent dye Antonia Red-Dextran (3000 MW, Sigma-Aldrich) and N/OFQ (MedChem Express) were simultaneously applied in bolus to sensor-expressing HEK293T cells at 37 °C with a stage-top incubator (Tokai Hit). Fluorescent signals were excited at 488 nm (NOPLight) and 561 nm (Red-Dextran dye) and recorded using the high-speed line-scan function (Zeiss LSM 800) at 800 Hz. The onset latency of each experiment was first determined by calculating the time for the red-dextran fluorescent signal to reach 85% of the maximal value at the plateau. Only experiments with an onset latency smaller than 50 ms were considered in subsequent analysis to minimize the contribution of N/OFQ peptide diffusion to the temporal profile of sensor response. Membrane-corresponding pixels were first selected by thresholding pixel-wise ΔF/$F_0$ at 65% criteria. The fluorescent signal change of each membrane pixel was then normalized and fitted by a mono-exponential association model using a custom-written MATLAB script to derive the time constant $\tau_{on}$.

To obtain the time constant for ligand wash-off, HEK293T cells were cultured on poly-D-lysine (Gibco) coated 18-mm coverslips in Dulbecco's modified Eagle's medium (Gibco) with 10% fetal bovine serum (Gibco) and antibiotic-antimycotic (1:100 from 10,000 units/ml penicillin; 10,000 μg/ml streptomycin, 25 μg/ml amphotericin B, Gibco) mix at 37 °C and 5% $CO_2$. Cells were transduced at 70% confluency using the Effectene transfection kit (QIAGEN) and imaged after 24–48 h. On the day of recording, coverslips with NOPLight-expressing HEK293T cells were transferred to the imaging chamber perfused with 37 °C HBSS (Gibco). Fluorescence response was obtained with a blue (469 nm) LED (Colibri 7, Zeiss) on an upright Axio Examiner A1 microscope (Zeiss) using an N-Achroplan 10x/0.3 M27 objective (Zeiss). Images were collected at a sampling rate of 1 Hz (Live Acquisition, Thermo Fisher Scientific). For wash-off experiments, N/OFQ was

added into the imaging chamber to a final concentration of 500 nM with the perfusion flow turned off. The perfusion system was turned on after 30 s of ligand application to allow equilibrium of sensor response. For puffed application experiments, a glass pipette mounted on a microinjector (Nanoject II™ Drummond) filled with N/OFQ was placed near the imaged field of view. About 50 nL of 100 μM N/OFQ was puffed onto the cells at a rate of 50 nL/s. Regions of interest for quantification of the fluorescence response was selected in Fiji (ImageJ). All cell membranes in the field of view were selected for analysis for wash-off experiments, whereas in puffed application experiments, only cell membranes within the radius of ½ of the glass pipette diameter were selected. A mono-exponential decay model was fitted to the data using Curve Fitting Toolbox (Matlab) to deduce the time constant $\tau_{off}$.

## One-photon spectral characterization
One-photon fluorescence excitation ($l_{em} = 560$ nm) and emission ($l_{exc} = 470$ nm) spectra were determined using a Tecan M200 Pro plate reader at 37 °C. HEK293T cells were transfected with an Effectene transfection kit (QIAGEN). Twenty-four hours after transfection, cells were dissociated with Versene (Thermo Fisher) and thoroughly washed with PBS. Next, cells were resuspended in PBS to a final concentration of $3.3 \times 10^6$ cells/mL and aliquoted into two individual wells of a 96-well microplate with or without ligand (N/OFQ or Ro 64-6198, 1 μM), together with two wells containing the same amount of non-transfected cells to account for autofluorescence and a single well containing PBS to determine the Raman bands of the solvent. To determine the excitation and emission spectra of NOPLight at various pH, PBS was adjusted to the desired pH using 10 M HCl or 10 M NaOH.

## Two-photon brightness characterization
Two-photon brightness profiles of NOPLight were obtained from HEK293T cells before and after the addition of N/OFQ (1 μM). Cells were transfected with Lipofectamine 3000 and were imaged 24 h post-transfection. The medium was replaced with PBS prior to imaging in order to avoid DMEM autofluorescence. The two-photon spectra were acquired as described previously[45].

## cAMP assay
HEK293 cells growing at 70% confluency in a 10-cm dish were transfected with wild-type human NOPR or NOPLight (3 μg DNA) and GloSensor-20F (Promega, 2.5 μg DNA) using 12 μL Lipofectamine 2000 (Thermo Fisher) as in ref. 73. After 24 h, cells were plated into clear-bottom 96-well plates at 200,000 cells/well in DMEM (without phenol red, with 30 mM HEPES, p.H. 7.4) containing 250 μg/ml luciferin and incubated for 45–60 min at 37 °C. Cells were treated with forskolin (3 μM) and varying concentrations of N/OFQ immediately followed by image acquisition every 45 s for 30 min at 37 °C using a Hidex Sense plate reader. Luminescence values were normalized to the maximum luminescence values measured in the presence of 3 μM forskolin and to vehicle-treated control cells.

## TIRF microscopy
HEK293 cells growing on polylysine-coated 35-mm, glass-bottom dishes (MatTek, P35G-1.5-14-C) were transfected with wild-type human NOPR or NOPLight (0.8 μg DNA) and β-arrestin-2-mCherry (1 μg DNA) using 3 μL Lipofectamine 2000 (Thermo Fisher). After 24 h, receptors were surface-labeled for 10 min with anti-FLAG M1-AF647[45] and media changed to HBS imaging solution (Hepes buffered saline (HBS) with 135 mM NaCl, 5 mM KCl, 0.4 mM MgCl₂, 1.8 mM CaCl₂, 20 mM Hepes, 5 mM D-glucose adjusted to pH 7.4). Cells were imaged at 37 °C using a Nikon TIRF microscope equipped with a 100×1.49 oil CFI Apochromat TIRF objective, temperature chamber, objective heater, perfect focus system, and an Andor DU897 EMCCD camera, in time-lapse mode with 10 s intervals. The laser lines used were 561 nm

(for β-arrestin-2) and 647 nm (for receptor constructs). 10 μM N/OFQ was added by bath application. Protein relocalization (ΔF) was calculated as F(t)/F0 with F(t) being the β-arrestin-2 signal at each time point (t) (normalized to M1-AF647 signal, when specified) and F0 being the mean signal before ligand addition.

## Virus production
The adeno-associated virus (AAV) encoding NOPlight was produced by the viral vector facility at the University of Zurich (VVF). The AAVs encoding the NOPlight-ctr sensor and the Cre-dependent NOPlight were produced by Vigene Biosciences. All other viruses used in this study were obtained either from the VVF or Addgene. The viruses used in this study were: AAVDJ-hSyn-NOPLight, $4.1 \times 10^{13}$ GC/ml; AAVDJ-hSyn-NOPlight-ctr, $2.9 \times 10^{13}$ GC/ml; AAVDJ-hSyn-FLEX-NOPlight, $2.5 \times 10^{13}$ GC/ml; AAV5-EF1a-DIO-HA-hM3D(Gq)-mScarlet, $1.7 \times 10^{13}$ GC/ml; AAV5-DIO-hSyn-mCherry, $2.4 \times 10^{12}$ GC/ml; AAV8-hSyn-DIO-hM3Dq.mcherry, $1 \times 10^{13}$ GC/ml; AAV-DJ-EF1a_DIO-GCaMP6m $1.2 \times 10^{13}$ GC/ml; AAV1-hSyn1-NES-jRCaMP1b, $6.9 \times 10^{12}$ GC/ml.

## Animals
Ten- to 24-week-old, wild-type C57BL/6 mice, PNOC-Cre (as described previously[22,23]) and OPRL1-Cre mice were used in this study. Rat embryos (E17) obtained from timed-pregnant Wistar rats (Envigo) were used for preparing primary cortical neuronal cultures. Animal procedures were performed in accordance to the guidelines of the European Community Council Directive or the Animal Welfare Ordinance (TSchV 455.1) of the Swiss Federal Food Safety and Veterinary Office and were approved by the Zürich Cantonal Veterinary Office, and the other respective local government authorities (Bezirksregierung Köln; Animal Care and Use Committee of University of Washington). Mice were housed in the animal facility at 22–24 °C on a 12 h/12 h reverse light/dark cycle (7:00 AM lights off) in ventilated cages receiving standard chow and water access ad libitum. Animals placed in the Pavlovian and operant conditioning paradigms were food-restricted down to ~90% of their ad libitum body weight beginning 1 week prior to conditioning for the entire duration of the paradigm.

## Stereotaxic surgery
**Viral vector injections and optic fiber implantation in the arcuate nucleus (ARC).** All surgeries in the arcuate nucleus were performed on male adult mice aged 8–10 weeks. Animals were anesthetized with 5% isoflurane and maintained at 1.5–2% throughout the surgery. For cell-specific DREADD expression, adeno-associated virus encoding AAV8-hSyn-DIO-hM3Dq.mcherry (Addgene #44361, 100 nL, viral titer $1 \times 10^{13}$ GC/mL) was injected using a glass capillary into the ARC (−1.5 mm AP, −0.3 mm ML, −5.78 mm DV). Adeno-associated virus encoding AAV-DJ-hSyn-NOPLight (500 nL, viral titer $0.8 \times 10^{13}$ GC/mL) was injected into the ARC in either wild-type or DREADD-expressing PNOC-Cre mice. All viruses were injected at a rate of 100 nL/min. For in vivo fiber photometry, after 5 min of viral injections a sterile optic fiber was implanted (diameter 400 μm, Doric Lenses) at the following coordinates (−0.450 mm AP, −0.2 mm ML, −5.545 mm DV) at an 8° angle. The implant was fixed with dental acrylic and closed with a cap after drying. Mice were allowed to recover from surgery 4 weeks before in vivo fiber photometry recordings started.

**Viral vector injections and optic fiber implantation in the VTA.** Following a minimum of 7 days of acclimation to the holding facility, mice were initially anesthetized in an induction chamber (1–4% isoflurane) and placed into a stereotaxic frame (Kopf Instruments, Model 1900) where anesthesia was maintained at 1–2% isoflurane. Depending on the specific experimental paradigm, mice received viral injections either unilaterally or bilaterally using a blunt neural syringe (86200, Hamilton Company) at a rate of 100 nL/min. For exogenous pharmacology experiments, wild-type (WT) mice were injected with AAV-DJ-hSyn-

NOPLight (200–500 nL, viral titer $2$–$4 \times 10^{12}$ vg/mL) or AAV-DJ-hSyn-NOPLight-ctr (300 nL, viral titer $2.9 \times 10^{13}$ vg/mL) and OPRL1-Cre mice were injected with AAV-DJ-hSyn-FLEX-NOPLight (300 nL, viral titer $2.5 \times 10^{13}$ vg/mL). For chemogenetic experiments, PNOC-Cre mice were co-injected with a 1:1 mix of AAV-DJ-hSyn-FLEX-NOPLight (300 nL, viral titer $2.5 \times 10^{12}$ vg/mL) and either AAV5-EF1a-DIO-HA-hM3D(Gq)-mScarlet (300 nL, viral titer $1.7 \times 10^{13}$ vg/mL) or AAV5-DIO-hSyn-mCherry (300 nL, viral titer $2.4 \times 10^{12}$ vg/mL). For head-fixed behavioral experiments, WT mice were injected with AAV-DJ-hSyn-NOPLight (300 nL, viral titer $4.1 \times 10^{12}$ vg/mL). For freely moving tail lift experiments, PNOC-Cre mice were injected with AAV-DJ-EF1a-DIO-GCaMP6m (300 nL, viral titer $1.2 \times 10^{13}$ vg/mL); OPRL1-Cre mice were injected with AAV-DJ-hSyn-FLEX-NOPLight (300 nL, viral titer $2.5 \times 10^{12}$ vg/mL); WT mice were injected with AAV-DJ-hSyn-NOPLight-ctr (300 nL, viral titer $2.9 \times 10^{12}$ vg/mL). For freely moving reward-seeking experiments (Pavlovian conditioning, operant conditioning, progressive ratio test), WT mice were injected with AAV-DJ-hSyn-NOPLight (300 nL, viral titer $4.1 \times 10^{13}$ vg/mL). All viruses were injected into the VTA (stereotaxic coordinates from Bregma: −3.3 to −3.4 AP, +1.6 ML, −4.75 to −4.3 DV) at a 15° angle, with an optic fiber implanted above the injection site. For animals undergoing head-fixed behaviors, a stainless-steel head-ring was also secured to allow for head-fixation. All implants were secured using Metabond (C & B Metabond). Animals were allowed to recover from surgery for a minimum of 3 weeks before any behavioral testing, permitting optimal viral expression.

**Viral vector injections in the nucleus accumbens.** Surgeries were performed on wild-type mice (male and female) aged 8–10 weeks. Anesthesia was induced using 4–5% isoflurane and maintained at 2%. AAV-DJ-hSyn-NOPLight (300 nL, viral titer $4 \times 10^{12}$ vg/mL) was injected bilaterally (Nanoject II™ Drummond) at +1.5 AP, ±0.7 ML, −4.5 DV. Animals were allowed to recover from surgery for a minimum of 4 weeks before the experiment, permitting optimal viral expression.

**Acute brain slice preparation, electrophysiology, and imaging.** For recordings in the ARC, experiments were performed at least 4 weeks after viral injections. The animals were lightly anesthetized with isoflurane (B506; AbbVie Deutschland GmbH and Co KG, Ludwigshafen, Germany) and decapitated. Coronal slices (280 μm) containing NOPLight expression in the arcuate nucleus (ARC) were cut with a vibration microtome (VT1200 S; Leica, Germany) under cold (4 °C), carbogenated (95% $O_2$ and 5% $CO_2$), glycerol-based modified artificial cerebrospinal fluid (GaCSF: 244 mM Glycerol, 2.5 mM KCl, 2 mM MgCl₂, 2 mM CaCl₂, 1.2 mM NaH₂PO₄, 10 mM HEPES, 21 mM NaHCO₃, and 5 mM Glucose adjusted to pH 7.2 with NaOH). The brain slices were continuously perfused with carbogenated artificial cerebrospinal fluid (aCSF: 125 mM NaCl, 2.5 mM KCl, 2 mM MgCl₂, 2 mM CaCl₂, 1.2 mM NaH₂PO₄, 21 mM NaHCO₃, 10 mM HEPES, and 5 mM Glucose adjusted to pH 7.2 with NaOH) at a flow rate of ~2.5 ml/min. To reduce GABAergic and glutamatergic synaptic input, $10^{-4}$ M PTX (picrotoxin, P1675; Sigma-Aldrich), $5 \times 10^{-6}$ M CGP (CGP-54626 hydrochloride, BN0597, Biotrend), $5 \times 10^{-5}$ M DL-AP5 (DL-2-amino-5-phosphono-pentanoic acid, BN0086, Biotrend), and $10^{-5}$ M CNQX (6-cyano-7-nitroquinoxaline-2,3-dione, C127; Sigma-Aldrich) were added to perfusion aCSF. The imaging setup consisted of a Zeiss AxioCam/MRm CCD camera with a $1388 \times 1040$ chip and a Polychromator V (Till Photonics, Gräfelfing, Germany) coupled via an optical fiber into the Zeiss Axio Examiner upright microscope. NOPLight fluorescence was collected at 470 nm excitation with 200 ms exposure time and a frame rate of 0.1 Hz. The emitted fluorescence was detected through a 500–550 nm bandpass filter (BP525/50), and data were acquired using $5 \times 5$ on-chip binning. Images were recorded in arbitrary units (AU) and analyzed as 16-bit grayscale images. To investigate the responses of NOPLight to N/OFQ (cat# 0910, Tocris), increasing concentrations (5 nM, 10 nM, 50 nM, 100 nM, 500 nM, 1 μM, 5 μM, 10 μM) of N/OFQ

were sequentially bath-applied for 10 min each. Changes in fluorescent intensity upon ligand application were quantified by comparing the averaged fluorescent intensity measured in 3-min intervals immediately before and at the end of ligand application. For electrophysiological recordings, the preparation of the brain slices and the recording conditions were the same as for the imaging experiments. Perforated patch-clamp recordings were performed as previously described[23]. To investigate NOPLight response to chemogenetically evoked endogenous N/OFQ release, hM3Dq was activated by bath application of 3 μM clozapine-N-oxide (CNO, ab141704, Abcam) for 10 min. N/OFQ and CNO were bath-applied at a flow rate of ~2.5 ml/min. The analysis was performed offline using ImageJ (version 2.3.0/1.53 f) and Prism 9 (GraphPad, CA, USA). Amplitudes and kinetics of the signals were calculated as means (in AU) of fluorescent regions in the ARC, which were defined as the respective regions of interest (ROI, 0.15–0.2 mm²). Biexponential fits of the signals' time courses before the N/OFQ application were used to correct for bleaching.

For recordings in the NAc, experiments were performed at least 4 weeks after bilateral viral injections. Mice were anesthetized with an intraperitoneal injection of pentobarbital (200 mg/kg, 10 mL/kg) and decapitated. The brain was quickly extracted while submerged in ice-cold aCSF (120 mM NaCl, 2.5 mM KCl, 1.25 mM NaH₂PO₄, 26 mM NaHCO₃, 5 mM HEPES, 1 mM MgCl₂, 14.6 mM D-glucose, and 2.5 mM CaCl₂ at 305–310 mOsm/kg) bubbled with 95/5% $O_2/CO_2$. About 275-μm thick coronal slices containing the NAc were obtained using a vibratome (HM 650 V, Thermo Fisher Scientific). The slices were incubated at 34 °C for 20 min in continuously oxygenated aCSF. Following incubation, brain slices were transferred at RT and kept until recording. Recordings were conducted in a slice chamber kept at 31 °C perfused with aCSF. To visualize the NOPLight signal, slices were illuminated with a blue (469 nm) LED (Colibri 7, Zeiss) on an upright Axio Examiner A1 microscope (Zeiss) using an N-Achroplan 10x/0.3 M27 objective (Zeiss). Images were collected at a sampling rate of 1 Hz (Live Acquisition, Thermo Fisher Scientific). To locally puff N/OFQ, a glass pipette filled with the desired concentration of N/OFQ was mounted to a microinjector (Nanoject II™ Drummond) and positioned into the imaged field of view on the slice. Various concentrations of N/OFQ (in 50 nL) were puffed onto the slice at a 50 nL/s rate. Fluorescence responses were quantified in Fiji (ImageJ) by selecting a circular region of interest with a radius of ½ of the size of the glass pipette diameter.

## Tissue preparation and immunohistochemistry (IHC)
**IHC in the ARC.** Brain slices were fixed in Roti-Histofix (PO873, Carl Roth) for ~12 h at 4 °C and subsequently rinsed in 0.1 M phosphate-buffered saline (PBS, $3 \times 10$ min). PBS contained (in mM) 72 Na₂HPO₄ x dihydrate, 28 NaH₂PO₄ monohydrate, resulting in pH 7.2. To facilitate antibody penetration and prevent unspecific antibody binding, brain slices were preincubated in PBS containing 1% (w/v) Triton X-100 (TX, A1388, AppliChem) and 10% (v/v) normal goat serum (NGS, ENG9010-10, Biozol Diagnostica) for 30 min at room temperature (RT). Brain slices were then incubated for ~20 h at RT with primary antibodies (chicken anti-GFP, 1:1000, ab13970, Abcam; rat anti-mcherry, 1:1000, Thermo Fisher Scientific, M11217) in PBS-based blocking solution containing 0.1% TX, 10% NGS and 0.001% sodium azide (S2002, Sigma-Aldrich). Brain slices were rinsed first in PBS-0.1% TX ($2 \times 10$ min, RT), then in PBS ($3 \times 10$ min, RT) and subsequently incubated with secondary antibodies (goat anti-chicken-FITC, Jackson #103-095-155, 1:500; goat anti-rabbit Alexa-Fluor-594, Thermo Fisher Scientific, A11012; 1:500) and DAPI (1:1000) for 2 h at room temperature. Brain slices were then rinsed in PBS-0.1% TX ($2 \times 10$ min, RT) and PBS ($3 \times 10$ min, RT), dehydrated in an ascending ethanol series, cleared with xylene (131769.1611, AppliChem), and mounted for imaging.

**IHC in the pnVTA.** Animals were transcardially perfused with 0.1 M phosphate-buffered saline (PBS) followed by 40 mL 4%

paraformaldehyde (PFA). Brains were extracted and post-fixed in 4% PFA overnight and then transferred to 30% sucrose in PBS for cryo-protection. Brains were sectioned at 30 μm on a microtome and stored in a 0.1 M phosphate buffer at 4 °C prior to immunohistochemistry. For behavioral cohorts, viral expression and optical-fiber placements were confirmed before inclusion in the presented datasets. Immunohistochemistry was performed as previously described[74,75]. In brief, free-floating sections were washed in 0.1 M PBS for 3 × 10 min intervals. Sections were then placed in a blocking buffer (0.5% Triton X-100 and 5% natural goat serum in 0.1 M PBS) for 1 h at room temperature. After blocking buffer, sections were placed in primary antibody (chicken anti-GFP, 1:2000, Abcam) overnight at 4 °C. After 3 × 10 min 0.1 M PBS washes, sections were incubated in secondary antibody (Alexa-Fluor 488 goat anti-chicken, Abcam) for 2 h at room temperature, followed by another round of washes (3 × 10 min in 0.1 M PBS, 3 × 10 min in 0.1 M PBS). After immunostaining, sections were mounted and coverslipped with Vectashield HardSet mounting medium containing DAPI (Vector Laboratories) and imaged on a Leica DM6 B microscope.

### Generation of OPRL1-Cre mouse line and reporter crosses

OPRL1[iresCre:GFP] knock-in mice were generated at the University of Washington. A cassette encoding IRES-mnCre:GFP was inserted just 3′ of the termination codon in the last coding exon of the *Oprl1* gene. The 5′ arm (12 kb with *Pac*I and *Sal*I sites at 5′ and 3′ends, respectively) and 3′ arm (3.5 kb with *Xho*I and *Not*I sites at 5′and 3′ ends, respectively) of the OPRL1 gene were amplified from a C57BL/6 BAC clone by PCR using Q5 Polymerase (New England Biolabs) and cloned into poly-linkers of a targeting construct that contained IRES-mnCre:GFP, an frt-flanked Sv40Neo gene for positive selection, and HSV thymidine kinase and *Pgk*-diphtheria toxin A chain genes for negative selection. The IRES-mnCre:GFP cassette has an internal ribosome entry sequence (IRES), a myc-tag, and nuclear localization signals at the N-terminus of Cre recombinase, which is fused to green fluorescent protein followed by an SV40 polyadenylation sequence (Cre:GFP). The construct was electroporated into G4 ES cells (C57BL/6 × 129 Sv hybrid) and correct targeting was determined by Southern blot of DNA digested with *BamH*I using a 32P-labeled probe downstream of the 3′ arm of the targeting construct. Five of the 77 clones analyzed were correctly targeted. One clone that was injected into blastocysts resulted in good chimeras that transmitted the targeted allele through the germline. Progeny were bred with *Gt(Rosa)26Sor*-FLP recombinase mice to remove the frt-flanked SV-Neo gene. Mice were then continuously backcrossed to C57Bl/6 mice.

To visualize OPRL1-Cre expression in the VTA, OPRL1-Cre mice were crossed to the Ai3 EYFP flox-stop reporter line (Jackson Lab, #007903). Adult OPRL1-Cre x Ai3 mice were transcardially perfused with 0.1 M phosphate-buffered saline (PBS) followed by 40 mL of 4% paraformaldehyde (PFA). Brains were extracted and post-fixed in 4% PFA overnight and then transferred to 30% sucrose in PBS for cryo-protection. Brains were sectioned at 30 μm on a microtome and stored in a 0.1 M phosphate buffer at 4 °C. Midbrain sections were mounted on Super Frost Plus slides (Thermo Fisher) and coverslipped with Vectashield HardSet mounting medium containing DAPI (Vector Laboratories) and imaged on a Leica DM6 B microscope.

### RNAscope fluorescent in situ hybridization (FISH)

Immediately after the rapid decapitation of OPRL1-Cre mice ($n = 2$), brains were extracted, flash frozen in −50 °C 2-methylbutane, and then stored at −80 °C. Brains were sectioned coronally into 15 μm slices on a cryostat at −20 °C, mounted onto Super Frost Plus slides (Fisher), and then stored at −80 °C prior to RNAScope FISH. FISH was performed according to the RNAScope Fluorescent Multiplex Assay for use with fixed frozen tissues (Advanced Cell Diagnostics, Inc.). Slides with 2–4 brain sections each that contained the VTA were post-fixed in

prechilled 10% neutral-buffered formalin for 15 min at 4 °C, dehydrated in ethanol, and then treated with a protease IV solution for 30 min at 40 °C. The sections were then incubated for 2 h at 40 °C with target probes (Advance Cell Diagnostics, Inc.) for mouse Oprl1 (accession number NM_011012.5, target region 988 – 1937), Th (accession number NM_009377.1, target region 483–1603), and Cre (target region 2–972). Next, sections underwent a series of probe amplification (AMP1-4) at 40 °C, including a final incubation with fluorescently labeled probes (Alex 488, Atto 550, Atto 647) targeted to the specified channels (C1–C3) that were associated with each of the probes. Finally, sections were stained with DAPI and slides were coverslipped with Vectashield HardSet mounting medium (Vector Laboratories). Slides were imaged on a Leica TCS SPE confocal microscope (Leica) at 60x magnification, and Fiji and HALO software were used to process images and quantify expression. Images were obtained under consistent threshold and exposure time standards. Leica images were opened and converted to TIFs for compatibility with HALO software. In HALO, cell ROIs were first made using the DAPI channel, and then Oprl1+ and Cre+ cells were identified if a fluorescent threshold for each channel was met within the cell ROI. Two to three separate slices were quantified for each animal.

### Photometry recording

**Recordings in ARC**. For fiber-photometry studies in the arcuate nucleus, the set-up of the photometry recorder[76] consisted of an RZ5P real-time processor (Tucker-Davis Technologies) connected to a light source driver (LED Driver; Doric Lenses). The LED Driver constantly delivered excitation light at 405 nm (control) and 465 nm (NOPLight) wavelengths. The light sources were filtered by a four-port fluorescence minicube (FMC_AE(405)_E1(460-490)_F1(500–550)_S, Doric Lenses) before reaching the animal. The fluorescence signals were collected from the same fiber using a photoreceiver (Model 2151, New Focus), sent back to the RZ5P processor, and gathered by Synapse software (v.95-43718 P, Tucker-Davis Technologies).

**Recordings in pnVTA**. For fiber-photometry studies in the pnVTA, recordings were made continuously throughout the entirety of the pharmacology (30 min), chemogenetic (55 min), head-fixed sucrose (25 min), tail lift (20 min), and conditioned reward-seeking (60 min) sessions. Prior to recording, an optic fiber was attached to the implanted fiber using a ferrule sleeve (Doric, ZR_2.5). In pharmacology and conditioned reward-seeking experiments, a 531-Hz sinusoidal LED light (Thorlabs, LED light: M470F3; LED driver: DC4104) was bandpass filtered (470 ± 20 nm, Doric, FMC4) to excite NOPLight and evoke NOPR-agonist dependent emission while a 211-Hz sinusoidal LED light (Thorlabs, M405FP1; LED driver: DC4104) was bandpass filtered (405 ± 10 nm, Doric, FMC4) to excite NOPLight and evoke NOPR-agonist independent isosbestic control emission. In chemogenetics, head-fixed sucrose, and tail lift experiments, a 531-Hz sinusoidal LED light (Thorlabs, LED light: M490F3; LED driver: DC4104) was bandpass filtered (490 ± 20 nm, Doric, FMC6) to excite NOPLight and evoke NOPR-agonist dependent emission while a 211-Hz sinusoidal LED light (Doric, CLED_435; Thorlabs, LED driver: DC4104) was bandpass filtered (435 ± 10 nm, Doric, FMC6) to excite NOPLight and evoke NOPR-agonist independent isosbestic control emission. Prior to recording, a minimum 120 s period of NOPLight excitation with either 470-nm and 405-nm or 490-nm and 435-nm light was used to remove the majority of baseline drift. Power output for each LED was measured at the tip of the optic fiber and adjusted to ~30 μW before each day of recording. NOPLight fluorescence traveled back through the same optic fiber before being bandpass filtered (525 ± 25 nm, Doric, FMC4 or FMC6), detected with a photodetector system (Doric, DFD_FOA_FC), and recorded by a real-time processor (TDT, RZ5P). The 531-Hz and 211-Hz signals were extracted in real-time by the TDT program Synapse at a sampling rate of 1017.25 Hz.

**In vivo animal experiments.** All animal behaviors were performed within a sound-attenuated room maintained at 23 °C at least 1 week after habituation to the holding room. Animals were handled for a minimum of 3 days prior to experimentation, as well as habituated to the attachment of a fiber photometry patch cord to their fiber implants. For all experiments, mice were brought into the experimental room and allowed to acclimate to the space for at least 30 min prior to beginning any testing. All experiments were conducted in red light to accommodate the reverse light cycle schedule, unless otherwise stated. All pharmacological interventions were administered in a counterbalanced manner. All sessions were video recorded.

### In vivo pharmacology experiments in the ARC

Mice injected with AAV-DJ-hSyn-NOPLight were acclimatized to the behavior set-up 3 weeks post-surgery for 1 week. Awake animals were placed individually in a box, and an optic fiber cable was connected to the implanted fiber. The optic fiber was attached to a swivel joint above the box to avoid moving limitations. Recording started 5 min after optic fiber tethering. A 10-min-long baseline was recorded prior to the i.p. injection of vehicle or RO 64-6198 (v doses). Animals have no access to water or food during the recording session.

### In vivo pharmacology experiments in the pnVTA

WT mice injected with either AAV-DJ-hSyn-NOPLight ($n = 16$) or AAV-DJ-hSyn-NOPLight-ctr ($n = 6$) and OPRL1-Cre mice injected with AAV-DJ-hSyn-FLEX-NOPLight ($n = 3$) were allowed to recover a minimum of 3 weeks after surgery (adult male and female, 4–6 months old). Three days before testing they were habituated to handling, fiber photometry cable attachment, and to the behavioral test box. On test day, animals were placed into the behavioral test box, which was a 10" x 10" clear acrylic box with a layer of bedding on the floor illuminated by a dim, diffuse white light (~30 lux). Fiber photometry recordings were made using a 405 nm LED as the isosbestic channel and a 470 nm LED as the signal channel. After starting the photometry recording, the mice were free to move around the box with no intervention for 5 min to establish a baseline photometry signal. At 5 min into the recording, NOPLight mice ($n = 16$) were scruffed and received an intraperitoneal (i.p.) injection of vehicle or 1, 5, or 10 mg/kg of the selective NOPR agonist Ro 64-6198 and were recorded for an additional 25 min. NOPLight-ctr and FLEX-NOPLight mice received an i.p. injection of 10 mg/kg Ro 64-6198 5 min into the recording. Two subsets of the NOPLight animals were recorded on three separate, counterbalanced days (at least 24 h apart). The first subset ($n = 4$) received either (i) an i.p. injection of 10 mg/kg Ro 64-6198 5 min into the recording (RO), (ii) an oral gavage (o.g.) treatment with 10 mg/kg of selective NOPR antagonist LY2940094 5 min into the recording (LY), or iii) an o.g. treatment with 10 mg/kg LY2940094 30 min prior to the recording followed by an i.p. injection with 10 mg/kg Ro 64-6198 5 min into the recording (LY pretreatment + RO). The second subset ($n = 4-9$) received either (i) an i.p. injection of 10 mg/kg Ro 64-6198 5 min into the recording (RO), (ii) an i.p. injection of 10 mg/kg of selective NOPR antagonist J-113397 5 min into the recording (J11), or (iii) an i.p. injection of 10 mg/kg J-113397 30 min prior to the recording followed by an i.p. injection with 10 mg/kg Ro 64-6198 5 min into the recording (J11 pretreatment + Ro).

### In vivo chemogenetics (DREADD) experiments

PNOC-Cre mice co-injected with AAVDJ-hSyn-FLEX-NOPLight and either AAV5-EF1a-DIO-HA-hM3D(Gq)-mScarlet ($n = 8$) or AAV5-DIO-hSyn-mCherry ($n = 3$) were allowed to recover a minimum of 3 weeks after surgery (adult male, 4–6 months old). Three days before testing they were habituated to handling, fiber photometry cable attachment, and to the behavioral test box. On test day, animals were placed into the behavioral test box, which was a 10" x 10" clear acrylic box with a layer of bedding on the floor illuminated by a dim, diffuse white light

(~30 lux). Fiber photometry recordings were made using a 435 nm LED as the isosbestic channel and a 490 nm LED as the signal channel. After starting the photometry recording, the mice were free to move around the box with no intervention for 10 min to establish a baseline photometry signal. At 10 min into the recording, mice were scruffed and received an intraperitoneal (i.p.) injection of 5 mg/kg clozapine-N-oxide (CNO) and were recorded for an additional 45 min. A subset of the DIO-hM3D(Gq) animals ($n = 3$) were recorded on two separate, counterbalanced days (at least 24 h apart). On one recording day, they received the CNO treatment and recording timeline described above. On the other recording day, they were administered an oral gavage (o.g.) treatment with 10 mg/kg of the selective NOPR antagonist LY2940094 30 min prior to the photometry recording. The LY-pretreatment group then underwent the same recording timeline and CNO injection (5 mg/kg i.p., after a 10 min baseline) as the CNO-only day.

### Head-fixed cued-sucrose access paradigm

WT mice injected with AAV-DJ-hSyn-NOPLight ($n = 7$) and implanted with an optic fiber and stainless-steel head-ring to allow for head-fixation during fiber photometry recording were allowed to recover a minimum of 3 weeks after surgery (adult male, 4–6 months old). One week prior to behavioral testing, mice were food-restricted down to ~90% of their free-feeding body weight. For the 4 days prior to testing, animals were habituated to handling, fiber photometry cable attachment, and head-fixation. Animals were head-fixed to minimize motion-related artifacts in the fiber photometry signal, and all head-fixed testing was completed using the open-source OHRBETS platform[77].

Fiber photometry recordings of NOPLight signal during tone-cued access to 10% sucrose solution were made over two counterbalanced sessions where animals received an i.p. injection 30 min prior to the recording of either (i) 20 mg/kg NOPR antagonist J-113397 or (ii) vehicle. Each session consisted of a 5-min baseline period where animals were head-fixed with no stimuli delivery, 15 tone-cued-sucrose trials, and then a 5-min baseline period at the end of the session (25 min total). Each cued-sucrose trial consisted of a 5-s auditory tone (4 kHz, 80 dB) immediately followed by a 5-s extension of a retractable lick spout and delivery of five pulses of 10% sucrose solution (~1.5 μL/pulse, 200 ms inter-pulse interval) where mice could lick the spout to consume the solution. Cued-sucrose trials were separated by a variable inter-trial interval of 45 to 75 s.

All behavioral hardware was controlled using an Arduino Mega 2560 REV3 (Arduino) and custom Arduino programs. Individual licks were detected using a capacitive touch sensor (Adafruit MPR121) that was attached to the retractable lick spout. The pulsed sucrose delivery was controlled by a solenoid (Parker 003-0257-900). The timing of solenoid openings and lick events were recorded and synchronized with the photometry signal via TTL communication from the Arduino Mega to the fiber photometry system. Fiber photometry recordings were made using a 435 nm LED as the isosbestic channel and a 490 nm LED as the signal channel.

### Tail lift behavioral experiments

PNOC-Cre mice injected with AAV-DJ-EF1a-DIO-GCaMP6m ($n = 4$), OPRL1-Cre mice injected with AAV-DJ-hSyn-FLEX-NOPLight ($n = 3$), and WT mice injected with AAV-DJ-hSyn-NOPLight-ctr ($n = 3$) were allowed to recover a minimum of 3 weeks after surgery (adult male, 10–14 weeks old). Three days before testing they were habituated to handling, fiber photometry cable attachment, and to the behavioral test box. On test day, animals were placed into the behavioral test box, which was a 10" x 10" clear acrylic box illuminated by a dim, diffuse white light (~30 lux). Fiber photometry recordings were made using a 435-nm LED as the isosbestic channel and a 490-nm LED as the signal channel. After starting the photometry recording, mice were free to move around the box with no intervention for 5 min to establish a

baseline photometry signal. After the 5-min baseline, mice underwent four tail lift trials where they were suspended by the tail for 10 s and then gently returned to the behavioral test box. Tail lift trials were separated by a variable inter-trial interval of 120–300 s. All suspensions were made to the same height. After the final trial, photometry recording continued for an additional 5 min to establish a post-test signal baseline.

### Reward-seeking (Pavlovian, operant) conditioning paradigms

One week prior to Pavlovian conditioning and 3 weeks after surgery, WT fiber photometry mice expressing AAV-DJ-hSyn-NOPLight in the VTA ($n = 4$) were food-restricted down to ~90% of their free-feeding body weight (adult male, 4 months old). All reward-seeking training was completed in Med-Associates operant conditioning boxes (ENV-307A). Fiber photometry recordings were made using a 405 nm LED as the isosbestic channel and a 470 nm LED as the signal channel. Mice were first trained to associate illumination of a house light (CS, 5 s) with delivery of a single sucrose pellet (US) occurring immediately after the house light turned off. A randomized inter-trial interval of between 30 and 120 s separated consecutive trials. Pavlovian conditioning sessions lasted for 60 min, during which an average of 36–38 rewards were presented. Pavlovian conditioning was repeated over 5 days a total of five times, with simultaneous fiber-photometry recordings made during session 1 and session 4. Animals were then moved onto a fixed ratio 1 (FR1) schedule for 5 days (60 min/session), where they were required to perform a nose poke in the active nose poke port one time to receive the 5 s house light cue and subsequent pellet delivery. Pokes made into the inactive port had no effect. Simultaneous fiber-photometry recordings were made during FR1 sessions 1 and 4. Following FR1 training, the ratio was increased to an FR3 schedule for 4 days (60 min/session) requiring the mice to perform three active port nose pokes to receive the house light cue and a sucrose pellet, with simultaneous photometry recording during session 3. Last, mice were placed in a single, 120 min session on a progressive ratio schedule (PR) where the nose-poke criteria for each subsequent reward delivery followed the geometric progression $n_i = 5e^{i/5} - 5$ (1, 2, 4, 6, 9, 12…), increasing in an exponential manner.

### Data analysis for photometry recordings

**Recordings in ARC.** Fiber-photometry data was pre-processed by downsampling the raw data to 1 Hz and removing the first and last seconds of each recording to avoid noise. To correct for bleaching, the fluorescence decay as the baseline ($F_0$) for both signal and control channel were fitted using a Power-like Model[47]. If no model could be fitted for a sample (e.g., no decay), the median of the baseline recording was used as a substitute. The relative change post-injection ($\Delta F/F_0$) was estimated by ($\Delta F_t/F_0 = (F_t - F_0)/F_0$) for both the signal and control channel separately (where $F_t$ is the raw signal at time $t$). $\Delta F/F_{0control}$ was subsequently subtracted from ($\Delta F/F_{0signal}$) to correct for motion artifacts, obtaining a final estimate of the relative change in fluorescence intensity for each sample.

**Recordings in pnVTA.** Custom MATLAB scripts were developed for analysing fiber-photometry data in the context of mouse behavior. A linear least squares (LLS) fit was applied to the control signal (405 or 435 nm) to align and fit it to the excitation signal (470 or 490 nm). For pharmacology and chemogenetics experiments where the entire length of the recording was analysed to evaluate long-term changes in NOPLight fluorescence following drug injection, LLS fit was calculated using the recording's 'baseline' period that preceded the injection (5–10 min), and the fitted isosbestic signal was subtracted from the excitation signal to detrend bleaching and remove movement artifacts. To reduce high-frequency noise, data were down-sampled by a factor of 300. The processed fiber photometry trace was then smoothed across a rolling 10 s window, and z-scored relative to the mean and

standard deviation of the baseline period preceding drug injection (first 5 or 10 min of the recording for pharmacology and chemogenetics experiments, respectively).

For all behavioral experiments (head-fixed sucrose, tail lift, conditioned reward seeking) where short epochs were evaluated, decay from bleaching was first detrended by fitting a fourth-degree polynomial function to the raw signal and isosbestic traces, then dividing by the resulting curve. Next, the LLS fit of the isosbestic signal to the excitation signal was calculated over the entire session and the excitation signal was normalized by dividing the resulting fitted isosbestic signal. To reduce high-frequency noise, data were down-sampled by a factor of 100. The processed traces were then smoothed across a rolling 1 s window, extracted in windows surrounding the onset of relevant behavioral events (e.g., nose poke, cue onset, reward delivery, tail lift), z-scored relative to the mean and standard deviation of each event window, and then averaged. The post-processed fiber photometry signal was analysed in the context of animal behavior during Pavlovian conditioning and operant task performance.

### Statistics and reproducibility

All data were averaged and expressed as mean ± SEM unless specified otherwise. Statistical significance was taken as *$p < 0.05$, **$p < 0.01$, ***$p < 0.001$, and ****$p < 0.0001$, as determined by Mann–Whitney test, Wilcoxon test, two-way repeated-measures ANOVA followed by Bonferroni post hoc tests as appropriate. All $n$ values for each experimental group are described in the corresponding figure legend. All in vitro and ex vivo experiments were performed at least three times independently with similar results. For behavioral experiments, group size ranged from $n = 3$ to $n = 16$. Statistical analyses were performed in GraphPad Prism 9 (GraphPad, La Jolla, CA) and MATLAB 9.9 (The MathWorks, Natick, MA). No data were excluded from the analyses. Group allocations used in this study were randomly assigned to animals and/or cultured cells. The Investigators were not blinded to allocation during experiments and outcome assessment.

### Reporting summary

Further information on research design is available in the Nature Portfolio Reporting Summary linked to this article.

## Data availability

DNA and protein sequences for the sensors developed in this study have been deposited on the National Center for Biotechnology Information database (accession numbers OQ067483 and OQ067484) and are available in Supplementary Note 1. DNA plasmids used for viral production have been deposited both on the UZH Viral Vector Facility (https://vvf.ethz.ch/) and on Addgene (NOPLight: Addgene plasmid # 195578, [https://www.addgene.org/195578/]; NOPLight-Ctr: Addgene plasmid # 195579). Viral vectors can be obtained either from the Patriarchi laboratory, the UZH Viral Vector Facility, or Addgene. Raw data is available at https://doi.org/10.5281/zenodo.11149714 or by emailing the corresponding authors. Source data are provided with this paper.

## Code availability

Custom MATLAB code is available at https://github.com/patriarchilab/NOPLight.

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

## Acknowledgements

The results are part of a project that has received funding from the European Research Council (ERC) under the European Union's Horizon 2020 research and innovation program (Grant agreement No.s 891959 and 101016787 to T.P.; 742106 to J.C.B.). We also acknowledge funding from the University of Zürich Forschungskredit (FK-20-042, X.Z.), the Swiss National Science Foundation (Grant No. 310030_196455, T.P.; PCEFP3_181282, M.S.), the NIMH P50MH119467 (M.R.B.), the NIH/NIMH F31 F31DA059438-01A1 (C.S.), and the Deutsche Forschungsgemeinschaft (401832153, 431549029, and EXC 2030-390661388, P.K.). We would like to thank Jean-Charles Paterna and the Viral Vector Facility of the Neuroscience Center Zürich (ZNZ) for their kind help with virus production.

## Author contributions

T.P. and M.R.B. led the study. X.Z., C.S., T.P., and M.R.B. conceptualized and designed the study. X.Z. developed the NOPLight and NOPLight-ctr sensors, performed in vitro sensor screening and characterization, including confocal imaging in HEK293T cells and neurons, and performed kinetic measurements in acute brain slices, under the supervision of T.P. M.A.B. prepared primary neuronal cultured under the supervision of D.B. J.D. performed two-photon spectral characterization under the supervision of L.R., T.P., and B.W. K.A. performed signaling assays under the supervision of M.S. D.F. and P.K. performed and analysed acute brain slice electrophysiology and imaging experiments. P.O.P., C.A.B., and L.S. performed and analysed in vivo photometry data in the arcuate nucleus under the supervision of J.C.B. C.S. performed and analysed in vivo photometry and chemogenetic experiments in the pnVTA under the supervision of M.R.B. A.L.P., A.S., R.D.P., A.S.A., J.C.J., and S.J. helped with surgeries and behavioral animal training for experiments in pnVTA under the supervision of M.R.B. T.P., M.R.B., X.Z., and C.S. wrote the manuscript with contributions from all authors.

## Competing interests

T.P. is a co-inventor on a patent application (PCT/US17/62993) related to the genetically encoded sensor technology described in this article. The remaining authors declare no competing interests.

## Additional information

[1]Institute of Pharmacology and Toxicology, University of Zürich, Zürich, Switzerland. [2]Neuroscience Center Zurich, University and ETH Zürich, Zürich, Switzerland. [3]Center for the Neurobiology of Addiction, Pain, and Emotion, University of Washington, Seattle, WA, USA. [4]Departments of Anesthesiology and Pharmacology and Bioengineering, University of Washington, Seattle, WA, USA. [5]Molecular and Cellular Biology, University of Washington School of Medicine, Seattle, WA, USA. [6]Max Planck Institute for Metabolism Research, Cologne, Germany. [7]Excellence Cluster on Cellular Stress Responses in Aging Associated Diseases (CECAD) and Center for Molecular Medicine Cologne (CMMC), University of Cologne, Cologne, Germany. [8]School of Applied Sciences, State University of Campinas (UNICAMP), Limeira, Sao Paulo, Brazil. [9]Institute of Zoology, Department of Biology, University of Cologne, Cologne, Germany. [10]Department of Cell Physiology and Metabolism, University of Geneva, Geneva, Switzerland. [11]Howard Hughes Medical Institute and Departments of Biochemistry and Genome Sciences, University of Washington, Seattle, WA 98195, USA. [12]Policlinic for Endocrinology, Diabetes and Preventive Medicine (PEDP), University Hospital Cologne, Cologne, Germany. [13]These authors contributed equally: Xuehan Zhou, Carrie Stine. ✉e-mail: mbruchas@uw.edu; patriarchi@pharma.uzh.ch

