## [Peer Review File · Nature Communications]

REVIEWER COMMENTS

Reviewer #1 (Remarks to the Author):

Reviewer Comments to Authors:

Endogenous opioid peptides, derived from four major precursors, are playing essential roles in regulating both physiological and pathological processes. Undoubtedly, a sensitive and reliable method for monitoring in vivo dynamics of opioid peptides is a critical step for understanding the function of opioids. In this manuscript, Zhou, Stine and colleagues generate a new GPCR (NOPR) based fluorescent sensor for imaging one type of opioids, named nociception opioid, in the brain. They did careful characterizations of the new sensor (NOPLight) in cultures, acute slices, and in vivo. Their data indicate that NOPLight is a promising tool for detecting the exogenous and endogenous nociception opioid, and has high potential to benefit and even advance research in the opioids field. Overall, the paper is well written and provides enough technical details. I will support the acceptance of this manuscript once the following issues are fully addressed.

Major points:

1. Line 141, regarding the sensor development process, I expect the amino acids between the GPCR and cpGFP in ICL3 are essential for the sensors' dynamic ranges. Are there any special reasons for targeting mutagenesis efforts focused on ICL2, other than ICL3?
2. In Figure 1f, the authors only shown the on rate of NOPLight, how about the off rate of the NOPLight? Carefully measuring the off rate of the new sensors is essential for the data interpretation. For most of the data, except for the data shown in Figure 5, off rates of NOPLight signal are very slow, which may because of the endogenous releasing kinetics of N/OFQ are slow or just because slow off rates of the sensor.
3. The authors clearly showed the NOPLight could achieve high temporal resolution for monitoring N/OFQ dynamics in vitro, ex vivo and in vivo, readers may also want to know whether the new opioid sensor could achieve the detection of N/OFQ with good spatial resolution in living mice. As the author described In Line 59-61, NOPR and preproN/OFQ (PNOC)-expressing neurons are highly enriched in the medial prefrontal cortex (mPFC), it will be great if the author could perform some 2p imaging in living mice.

4. Regarding the in vivo recording shown in Figure 5, I fully appreciated that the authors did carefully characterization regarding the isobaric point, however, readers may not be fully convinced by such a small in vivo signal. (1) At the 405 channels, it was difficult to see the increase signal via Supplementary Fig. 6; (2) If the data shown in Supplementary Fig. 7b truly reflected the N/OFG dynamics, readers would expect to see opposite signals recorded at 405 nm and 470 nm, according to the spectrum data shown in Supplementary Fig. 3a-b? To strengthen the in vivo recording data shown in Figure 5, several control experiments are essential: (1) Do similar experiments using NOPLight-ctrl; (2) to rule out the pH effect, the pH sensitivity along with excitation spectrum (especially at 405 nm and 470 nm) of NOPLight should be carefully examined.

Minor points:

1. Figure 1c, in HEK293T cells, a clear increasing peak is observed when adding J11, however, such a peak is not observed in neurons or in NOPLight-ctrl expressing cells, does this mean there is a special regulation in HEK293T cells or just an artifact?

2. Figure 2a, are these images are representative? The membrane trafficking of NOPLight seems not very good in the acute brain slices.

3. Figure 2g, what is the difference between the upper and bottom figures?

4. Supplementary Figure 4f, the timepoint for images should be labeled.

5. In the manuscript, the authors used a lot “showed no response to...” (e.g., Line 153 and Line 191), do they perform statistical analysis with “zero”? If not, using some modest words is more accurate.

6. Line 162-163, why the (apparent) affinity of NOPLight is one order of magnitude lower (compared with native receptors)? My understanding is that NOPLight can directly measure the conformational changes of GPCRs, however, previous methods are based on the downstream signaling assays, which is more sensitive because of the downstream signal amplifications. To faithfully compare the sensors' and native GPCR's affinity side by side, the authors should do direct binding assays (e.g., radiolabeled ligand binding).

7. Line 179-181, the maximum response is related to the concentration you use.

8. Line 185, the authors described that the peak Ex under one photon is 472 nm, however, according to the data shown in Supplementary Figure 3, the peak Ex might be longer?

9. The format should be consistent throughout the main text and figures, for example, there should be a space between the number and the unit.

Reviewer #2 (Remarks to the Author):

In this study, the authors generate a new biosensor for the peptide nociceptin/orphanin-FQ (N/OFQ) which they name NOPLight. They demonstrate the specificity of the sensor for N/OFQ and investigate sensor responsiveness to known agonists/antagonists of the nociceptin receptor. A major stated goal of this study was to generate a sensor for natural, endogenous N/OFQ dynamics. To test if this was achieved, the authors use in vivo photometry targeting the VTA region, a known site of N/OFQ production, to investigate endogenous release of N/OFQ. They show that artificial chemogenetic stimulation of local PNOC neurons is capable of driving release of N/OFQ as detected by NOPLight. They also report NOPLight responses during appetitive and aversive stimuli in awake mice. The interpretation of unusual signal dynamics in control excitation wavelengths, the lack of use of the non-responsive control NOPLight, and generally small amplitude dynamics make the measurements of natural N/OFQ hard to trust. As a result, this study demonstrates a novel biosensor of N/OFQ but falls short of demonstrating a sensor useful for detecting physiologically released N/OFQ.

1) My most pressing concern is with the interpretation of the photometry signals recording using 405nm. Given the authors evidence that the isosbestic point is closer to 440nm (Supp Fig 3) and therefore the 405nm excitation is left of the isosbestic point, one would expect the direction of change in fluorescence emission with 405nm excitation to be opposite that of 470nm. However, the authors note that in Supp Fig 6, the fluorescence increased in response to RO injection in both the 470 and 405 signals, but don't address how this is contrary to what would be expected (one would expect RO to increase 470 but decrease 405 signal if the isosbestic point is 440nm). The results in Supp Figures 6 & 7 that show that the change at 405nm is in the same direction as that at 470nm indicate that the change is caused by some common artefact such as motion. The use of 435nm in additional experiments does not resolve this as the same motion artefact is apparent (e.g. Supp Figure 7) just weaker. This result does not build confidence that the reported dynamics of naturally released N/OFQ are real.

2) Seems like interpretation in Supp Figure 6 and 7 are based on a single example mouse. Given this is major issue with the study, investigating the 405 and 435 signals across more than 1 mouse should be done to build any confidence in the results. In addition, all of the in vivo data is presented entirely qualitatively and it appears that no statistical quantification is performed.

3) The authors have nicely generated a control sensor that doesn't respond to ligand, but do not use this sensor to address artefact issues in the in vivo photometry recordings. This is important to both show the usefulness of using this control sensor and to strengthen any evidence that NOPLight can detect real, naturally released peptide.

4) Given comments #1 - 3, I am not convinced that this generation of NOPLight, as determined by the data presented in this manuscript, would be able to reliably detect natural dynamics in N/OFQ. However, the sensor does appear able to track chemogenetically-evoked release and agonists. As such, without adding further engineering of the sensor, I would suggest rewording the title and text to not overstate the ability of detecting natural N/OFQ dynamics and address this limitation in the discussion.

5) The authors characterized NOPLight onset dynamics but did not report offset measurement which are important for the community. Similar fast removal of ligand and measurement of signal decay would address this.

6) There are some major issues with Results section pointing to the correct Supplementary Figure 7 and 8 panels. Supp Fig 7a,b are never mentioned in the results. Supp Figure 7 c,d, are toe pinches but results section refers to them as reward experiments. Supp Figure 8 is supposed to be aversive data but is all appetitive data.

7) The figures legends need more clear description of what the sample size was as in current format it is left to the reader to guess. In legends for Figure 5, for example, is the mean and sem across mice and therefore the sample size is 4?

Reviewer #3 (Remarks to the Author):

Thank you for sending me this interesting piece of Internationally collaborative work to review. The N/OFQ-NOP system has wide applicability across a number of fields and as such Nature Comms is an appropriate vehicle for dissemination. The authors indicate "The functional relevance of N/OFQ action in the mammalian brain remains unclear due to a lack of high-resolution approaches to detect this neuropeptide with appropriate spatial and temporal resolution" This study describes NOPLight – a genetically encoded sensor that reports the release of N/OFQ "with unprecedented temporal resolution ex vivo and in vivo". My own group published a method to measure single cell release from isolated immune cells (DOI: 10.1371/journal.pone.0268868) but this is limited, cumbersome and can only be used in vitro. NOPLight is a paradigm shift by comparison with much wider applicability – critically for use in vivo and with behavioural correlates.

The paper is complex but well written and easy to follow. I have the following comments to make;

1. Potency values from log crc would be better expressed at pEC50 values. N/OFQ potency at the sensor is high but a little weaker than in CNS (Ferrara paper – by reference).
2. It is interesting UFP-101 displayed no residual agonist activity – presumably expression was not huge ?
3. In vitro conclusions “These results indicate that while NOPLight retains the ligand selectivity of its parent receptor, its cellular expression has a very low likelihood of interfering with intracellular signaling; thus the sensor can be meaningfully utilized in physiological settings.” are supported by the data.
4. Fig 2c does not saturate and looks very weak. Without the top 2 concentrations does the rest of the curve saturate ?
5. The system provides robust antagonist (LY and J) sensitive readout for the Roche agonist in vivo.
6. Chemogenetic activation of VTA neurons increased N/OFQ (measured with the biosensor) that was prevented by a NOP antagonist (LY).
7. Linking release to behaviour is one of the real strengths of this study and widens the appeal.
8. I have no comments on the structure of the figures.

In summary, this excellent piece of work with technical rigour and wide applicability receives by strongest support for publication. My comments should be viewed as very minor.

David Lambert, Leicester UK.

(For full disclosure – I have provided NO confidential comments to the editor)

Reviewer #4 (Remarks to the Author):

Zhou, Stine, et al. describe the engineering and characterization of a fluorescent biosensor for the opioid peptide nociceptin called NOPLight. The sensor was constructed by insertion of circularly permuted GFP into intracellular loop 3 of the human nociceptin receptor (NOPR), analogous to previously described cpGFP-GPCR sensors. By testing ~100 variants within the linkers and intracellular loop 2 they optimized the max fluorescence response in cell culture to $\Delta F/F$ of ~4. The response time, ligand affinity, and membrane trafficking of the sensor appear reasonable and they demonstrate that the sensor does not contribute to GPCR-mediated intracellular signaling. They additionally make a "dead" sensor by introducing mutations that block ligand binding. The authors characterize the performance of NOPLight in cell culture, acute brain slice, and in vivo in mouse brain in response to added nociceptin, receptor agonists and antagonists, chemogenetic circuit activation, and naturalistic stimuli during Pavlovian conditioning.

Generally speaking, the sensor appears to perform reasonably, especially in vitro, and the results are well described and presented. This tool will likely be useful for the community and I would recommend publication, although the demonstration of in vivo responses to naturalistic stimuli is not super convincing.

My primary concerns are as follows:

-No amino acid sequence of the sensor is described in the manuscript (that I saw), this should be added

-The in vivo responses (Fig 5) are quite small and noisy. They are likely real since there is a qualitative difference during the consumption phase between rewarded and non-rewarded trial, but these data are not terribly convincing. Indeed, the data did not appear to be completely reproducible when repeated using multiple wavelengths in the fiber photometry for normalization (supp fig 8). The "dead" sensor was not used in these experiments and could be a good control. Are all of the data in Fig 5 from n=4 mice as suggested in caption 5a? If not this should be stated.

We would like to thank all Reviewers for their positive and constructive feedback on our manuscript. We have addressed their concerns and performed several additional experiments, as detailed in our point-by-point responses below. The original Reviewer's comments are shown in *blue italics*. Cited text from the revised manuscript is shown here in *black italics*. We also marked in blue any major text changes in the revised version of our manuscript to enable transparency and ease of identification.

Reviewer #1 (Remarks to the Author):

Reviewer Comments to Authors:

Endogenous opioid peptides, derived from four major precursors, are playing essential roles in regulating both physiological and pathological processes. Undoubtedly, a sensitive and reliable method for monitoring in vivo dynamics of opioid peptides is a critical step for understanding the function of opioids. In this manuscript, Zhou, Stine and colleagues generate a new GPCR (NOPR) based fluorescent sensor for imaging one type of opioids, named nociception opioid, in the brain. They did careful characterizations of the new sensor (NOPLight) in cultures, acute slices, and in vivo. Their data indicate that NOPLight is a promising tool for detecting the exogenous and endogenous nociception opioid, and has high potential to benefit and even advance research in the opioids field. Overall, the paper is well written and provides enough technical details. I will support the acceptance of this manuscript once the following issues are fully addressed.

Major points:

1. Line 141, regarding the sensor development process, I expect the amino acids between the GPCR and cpGFP in ICL3 are essential for the sensors' dynamic ranges. Are there any special reasons for targeting mutagenesis efforts focused on ICL2, other than ICL3?

We appreciate this insightful comment by the Reviewer on the rationale that we followed during the sensor engineering process. Indeed, amino acid residues flanking cpGFP are well known to play a critical role in determining the sensor's dynamic range. To date different approaches to engineering GPCR-based sensors exist that follow different rationales. In this work we chose to keep the inner linkers identical to those that we previously optimized in the dopamine sensor dLight (Patriarchi et al, Science 2018). For clarity, these are the residues highlighted in bold here: LSS-**LI**-cpGFP-**NH**-DQL. Instead, we focused our mutagenesis and screening efforts in the amino acid sequences surrounding them. We chose this approach following the rationale that the inner linkers are already close to optimal for coupling the conformational activation of the receptor with the fluorescence state of cpGFP, as they had been extensively optimized previously. We added in our current screening mutagenesis of the ICL2, once again based on our prior experience. Indeed, we previously found that a single point mutation in the ICL2 of dLight could lead to a drastic improvement in the sensor's dynamic range. Motivated by this finding we tried the same approach on NOPLight and could indeed identify a mutation that improved sensor response (I156K). We now added a statement to the results section of the manuscript to make this rationale clearer to readers.

2. In Figure 1f, the authors only shown the on rate of NOPLight, how about the off rate of the NOPLight? Carefully measuring the off rate of the new sensors is essential for the data interpretation. For most of the data, except for the data shown in Figure 5, off rates of NOPLight signal are very slow, which may because of the endogenous releasing kinetics of N/OFQ are slow or just because slow off rates of the sensor.

We agree with the Reviewer that measuring the OFF kinetics of the sensor would add value for interpreting the NOPLight signals that we obtained both *ex vivo* and *in vivo*. Initially we attempted patch-clamp fluorometry (PCF) experiments, which have the advantage of very

rapid ligand removal by stepping into ligand-free solution (similar to e.g. Figure 1g-j in Kagiampaki Z. et al, Nat Methods 2023, PMID: 37474807). However, this technique relies on excised membrane patches, which typically limit the recording time to a few minutes (in conditions of high flow-speed). The slower ligand dissociation kinetics of NOPLight compared to monoamine sensors complicated these recordings, since the patch is typically pre-exposed to ligand solution during positioning of the patch-pipette, which required unreasonably long waiting times together with long interpulse intervals. Due to these limitations we were not able to obtain PCF data for NOPLight. Instead, we performed experiments in which cells were imaged under constant perfusion and the peptide ligand N/OFQ was either washed-on and washed-off from the perfusion system, or was directly applied locally onto the cells via a glass capillary positioned next to the cell membrane. From these two types of *in vitro* experiments we could measure OFF kinetics for NOPLight in the range of 60 sec (with perfused ligand experiments) or 30 sec (with local application of the ligand). These data are in line with our expectations based on previously reported kinetics of other peptide sensors, such as the oxytocin sensor MTRIA_{OT} ($\tau_{\text{OFF}} = 26$ sec, PMID: 36138174) or the CRF sensor GRAB_{CRF} ($\tau_{\text{OFF}} = 9.7$ sec, PMID: 37972184). To gain further insights on the kinetics of NOPLight in a more physiological environment, we also performed local application of N/OFQ in acute brain slices expressing the sensor. Also in this case we found the OFF-kinetics of the sensor to fall within the 30-60 sec range, in line with *in vitro* results. We have added these new findings to the results section of the manuscript (new **Figure 1g**, and new **Supplementary Figure 6**).

3. The authors clearly showed the NOPLight could achieve high temporal resolution for monitoring N/OFQ dynamics in vitro, ex vivo and in vivo, readers may also want to know whether the new opioid sensor could achieve the detection of N/OFQ with good spatial resolution in living mice. As the author described In Line 59-61, NOPR and preproN/OFQ (PNOC)-expressing neurons are highly enriched in the medial prefrontal cortex (mPFC), it will be great if the author could perform some 2p imaging in living mice.

While we agree with the Reviewer that *in vivo* two-photon experiments would be a nice addition to the story, these challenging experiments require dedicated expertise and sophisticated equipment that is not readily available to us at the moment. We would also like to clarify that, while it is true that NOPR and PNOC neurons are abundant in the mPFC, the behavioral conditions/stimuli that lead to N/OFQ release in the cortex are completely unknown. Thus, we feel that investigating these aspects would necessarily require a large-scale effort and result in a biological story of its own, falling beyond the scope of this tool development paper. Because of these reasons, unfortunately at the moment we cannot accommodate this request. It should also be considered that photometry-based applications of NOPLight such as the ones that we've showcased in the current article, can very well be leveraged by the community to empower new important discoveries involving N/OFQ physiology in the absence of an *in vivo* two-photon demonstration of the technique. Finally, we would like to note that other GPCR-based sensor tools have been previously published and have become extremely popular even in the absence of *in vivo* two-photon datasets as part of the initial tool-development paper (see for example: Feng et al, Neuron 2019, PMID: 30922875; Patriarchi et al, Nat Methods 2020, PMID: 32895537; Dong et al, Neuron 2023, PMID: 36924772; Quian et al, Nat Biotechnol, PMID: 36593404). To our knowledge there are few publications showing *in vivo* 2p imaging using neuropeptide sensors to date, and furthermore, given how little we know about nociceptin function in mPFC it would be an entire project to resolve. We definitely agree, however, that this is an important follow-up.

4. Regarding the in vivo recording shown in Figure 5, I fully appreciated that the authors did carefully characterization regarding the isobaric point, however, readers may not be fully convinced by such a small in vivo signal.

We agree that the behavioral *in vivo* signal we originally presented may not be convincing on its own. To address this, we evaluated NOPLight signal in two additional behaviors, including a tail suspension test where the z-scored change in fluorescence is ~5-fold greater than the

signal we originally reported (**Fig 5j-k**). We also completed additional control experiments using the NOPR antagonist (**Fig 5a-f**) and our control sensor NOPLight-ctr (**Fig 5j-k**). The signal we detected during a new head-fixed cued sucrose paradigm, while small, was completely blocked by pre-treatment with the NOPR antagonist J-113397 (**Fig 5c, f**). We also now demonstrate that NOPLight-ctr has no response during the tail suspension test, which is the behavior where we detected the largest fold-change in signal using NOPLight (**Fig 5j-k**). Lastly, we added rigorous statistical analyses on the originally presented *in vivo* data during Pavlovian and operant reward-seeking that emphasize statistically significant decreases in NOPLight signal during reward consumption (**Supplementary Fig 9, Fig 6**) as well as a significant increase in signal when animals expect to obtain a reward but do not (**Supplementary Fig 9i**).

(1) At the 405 channels, it was difficult to see the increase signal via Supplementary Fig. 6;

We agree that this increase in the 405 channel was difficult to see. The performance of the 405 nm isosbestic channel is challenging to present for *in vivo* photometry recordings which is why we originally chose to use representative traces, as the purpose of the 405 channel is to internally control for differences between photometry recordings that allows for averaging signal across recordings. We have removed this Supplementary figure in lieu of stronger findings from our new experiments where we thoroughly characterize the spectral properties and pH sensitivity of NOPLight and NOPLight-ctr at different wavelengths (described below).

(2) If the data shown in Supplementary Fig. 7b truly reflected the N/OFQ dynamics, readers would expect to see opposite signals recorded at 405 nm and 470 nm, according to the spectrum data shown in Supplementary Fig. 3a-b?

We agree that the data in this Supplementary figure regarding the isosbestic point may be confusing to readers. While the spectrum data would suggest that we would see opposite signals at 405 nm and 470 nm (**Supplementary Fig 3a-b**), there are other factors *in vivo* that can contribute to the fluorescence intensity, such as pH, which we have now characterized extensively (**Supplementary Fig 4**). Based on our new findings, one possibility explaining how the 405 and 470 channels could have signals in the same direction is that the behavior driving N/OFQ release may also cause the local pH to decrease, which would cause an increase in fluorescence at the 405 nm wavelength relative to its intensity at the higher pH (**Supplementary Fig 4c**). Furthermore, our new findings also indicate that this sensitivity to pH changes is more prominent at wavelengths closer to 405 nm than wavelengths closer to 435 nm, which could also explain why we observed smaller changes in the 435 nm channel than the 405 nm channel. However, we have ultimately removed this Supplementary figure to avoid this confusion in lieu of stronger *in vivo* control experiments completed during revision using the NOPR antagonist J113397 as well as the control sensor NOPLight-ctr (described in detail below).

To strengthen the in vivo recording data shown in Figure 5, several control experiments are essential:

(1) Do similar experiments using NOPLight-ctrl;

We generated new data using NOPLight-ctr demonstrating that it has no response during the tail suspension test, in contrast to FLEX-NOPLight which had a robust increase in signal for the duration of the tail suspension (**Fig 5j-k**). After calculating the area under the curve of these photometry traces over 5-second intervals, we found that NOPLight-ctr signal was unchanged from during tail suspension relative to baseline signal in the 5-second windows preceding and following the tail lift (**Fig 5k**). In contrast, FLEX-NOPLight signal was significantly elevated during tail suspension relative to its baseline levels preceding and following the tail lift, as well as relative to NOPLight-ctr signal during the tail lift (**Fig 5k**).

Notably, the tail suspension caused the largest fold-change in NOPLight signal, further demonstrating NOPLight-ctr's utility as a control for NOPLight signal *in vivo*.

(2) to rule out the pH effect, the pH sensitivity along with excitation spectrum (especially at 405 nm and 470 nm) of NOPLight should be carefully examined.

We agree with the Reviewer that it would be useful to have the additional characterization of NOPLight excitation at different pH levels. To address this, we did new experiments where we recorded the excitation and emission spectra of NOPLight in cells exposed to a range of extracellular pH from 6.0 to 8.0, similar to what was previously done by us and others in the field (Duffet L. et al, Nat Methods 2022, PMID: 35145320; Ino D. et al, Nat Methods 2022, PMID: 36138174). Our results, shown in Supplementary Figure 4, demonstrate that while the overall brightness of the sensor decreases as expected with decreasing pH, the isosbestic point remains within the wavelength range of 405-435 nm throughout the pH range.

Minor points:

1. Figure 1c, in HEK293T cells, a clear increasing peak is observed when adding J11, however, such a peak is not observed in neurons or in NOPLight-ctrl expressing cells, does this mean there is a special regulation in HEK293T cells or just an artifact?

The traces shown in Figure 1c are representative traces from single experiments. The peak observed when adding J11 onto HEK293T cells was due to imaging artifact since we manually add the antagonist via pipette application to the cells. We have replaced the trace with a more representative one.

2. Figure 2a, are these images representative? The membrane trafficking of NOPLight seems not very good in the acute brain slices.

The images in a are just an example of the brain slices used for *ex vivo* imaging. Please keep in mind that they have been processed for histology (PFA-fixation followed by anti-GFP labeling). This processing could sometimes affect the quality of apparent membrane trafficking given and can also highlight intracellularly-located sensors that are being trafficked to the surface. A second, better quality and more representative image of sensor expression in brain slices can be seen in panel e.

3. Figure 2g, what is the difference between the upper and bottom figures?

The upper and lower trace in the panel show the same data with a different alignment of the individual traces used for averaging. The trace in the upper panel is an average of individual traces aligned to the start of CNO application (physical switch of the perfused solution), while the trace in the lower panel is an average of individual traces aligned to the response onset (which accounts for natural variations in the delay of the response to CNO). We have clarified this in the figure legend.

4. Supplementary Figure 4f, the timepoint for images should be labeled.

We have now added timepoints to Supplementary Figure 4f.

5. In the manuscript, the authors used a lot "showed no response to..." (e.g., Line 153 and Line 191), do they perform statistical analysis with "zero"? If not, using some modest words is more accurate.

We agree with the Reviewer and have performed the necessary statistical analyses and revised the manuscript text accordingly.

6. Line 162-163, why the (apparent) affinity of NOPLight is one order of magnitude lower (compared with native receptors)? My understanding is that NOPLight can directly measure the conformational changes of GPCRs, however, previous methods are based on the

downstream signaling assays, which is more sensitive because of the downstream signal amplifications. To faithfully compare the sensors' and native GPCR's affinity side by side, the authors should do direct binding assays (e.g., radiolabeled ligand binding).

The Reviewer is correct in stating that GPCR-based sensors are less sensitive than other downstream signal assays because they only directly probe the conformational activation of the GPCR itself while the latter act via enzymatically amplified signaling cascades within the cells. In fact, we have also observed a higher potent effect of nociceptin at the wild-type receptor in our assay testing the intracellular signaling of this receptor. Being Gi-coupled, we used an assay in which cAMP is first induced via direct activation of Adenylate Cyclase (Forskolin) and then the ligand is titrated on the receptor-expressing cells. The data can be seen in Supplementary Figure 4a, from which we can see that the potency of nociceptin towards the wild-type receptor measured using this assay shows a half-maximal effect at a ligand concentration of around 3-4 nM (approximately 10-times higher than the apparent affinity we measured at NOPLight using its fluorescence as a readout). Our laboratory is unfortunately not equipped for radioligand binding assays, which required specialized equipment and training. Because of this, we regretfully cannot obtain such datasets. In any case, we feel that the additional information that these assays would provide would only add marginal value to the sensor characterization currently provided in the paper.

7. Line 179-181, the maximum response is related to the concentration you use.

The Reviewer is correct. We have rephrased this with:

“Interestingly, all of the agonist compounds tested induced an overall smaller fluorescence response than N/OFQ itself, when applied at the same concentration (Fig. 1g).”

8. Line 185, the authors described that the peak Ex under one photon is 472 nm, however, according to the data shown in Supplementary Figure 3, the peak Ex might be longer?

We would like to clarify that in the text the Reviewer is referring to we described “peak performance” of the sensor, which is the wavelength at which we observed the highest ratio between the fluorescent intensity measured in the presence and absence of N/OFQ (as represented by the grey dotted line in Supplementary Figure 3). On the other hand, the peak emission, which is the wavelength at which the emission of the sensor is greatest, is indeed found at a longer wavelength of 496 nm in the presence of N/OFQ. We now better clarified this in the results section of the manuscript.

9. The format should be consistent throughout the main text and figures, for example, there should be a space between the number and the unit.

We thank the Reviewer for pointing that out. We carefully proof-read the manuscript and fixed these issues to the best of our knowledge.

Reviewer #2 (Remarks to the Author):

In this study, the authors generate a new biosensor for the peptide nociceptin/orphanin-FQ (N/OFQ) which they name NOPLight. They demonstrate the specificity of the sensor for N/OFQ and investigate sensor responsiveness to known agonists/antagonists of the nociceptin receptor. A major stated goal of this study was to generate a sensor for natural, endogenous N/OFQ dynamics. To test if this was achieved, the authors use in vivo photometry targeting the VTA region, a known site of N/OFQ production, to investigate endogenous release of N/OFQ. They show that artificial chemogenetic stimulation of local PNOC neurons

is capable of driving release of N/OFQ as detected by NOPLight. They also report NOPLight responses during appetitive and aversive stimuli in awake mice. The interpretation of unusual signal dynamics in control excitation wavelengths, the lack of use of the non-responsive control NOPLight, and generally small amplitude dynamics make the measurements of natural N/OFQ hard to trust. As a result, this study demonstrates a novel biosensor of N/OFQ but falls short of demonstrating a sensor useful for detecting physiologically released N/OFQ.

1) My most pressing concern is with the interpretation of the photometry signals recording using 405nm. Given the authors evidence that the isosbestic point is closer to 440nm (Supp Fig 3) and therefore the 405nm excitation is left of the isosbestic point, one would expect the direction of change in fluorescence emission with 405nm excitation to be opposite that of 470nm. However, the authors note that in Supp Fig 6, the fluorescence increased in response to RO injection in both the 470 and 405 signals, but don't address how this is contrary to what would be expected (one would expect RO to increase 470 but decrease 405 signal if the isosbestic point is 440nm). The results in Supp Figures 6 & 7 that show that the change at 405nm is in the same direction as that at 470nm indicate that the change is caused by some common artefact such as motion. The use of 435nm in additional experiments does not resolve this as the same motion artefact is apparent (e.g. Supp Figure 7) just weaker. This result does not build confidence that the reported dynamics of naturally released N/OFQ are real.

We agree with the Reviewer that according to our *in vitro* characterization NOPLight shows decreased fluorescence emission when excited at 405 nm in the presence versus absence of N/OFQ. We further investigated the sensor's photophysical properties by obtaining Excitation and Emission spectra in the presence/absence of RO-64 agonist compound and could confirm a similar Ro-64-dependent decrease in sensor excitation at 405 nm. With this in mind, we would like to point out that the data presented in Supplementary Figure 6 were obtained *in vivo* from animals that received systemic administration of RO-64 (i.p.). The drug RO 64-6198 produces certain side effects including hypolocomotion and hypothermia (Shoblock et al, 2007 CNS Drug Rev., PMID: 17461893) which in turn are likely to cause a signal artifact under the recording fiber, given the known contamination of hemodynamics to fiber photometry signals (Zhang WT et al, 2022 Cell Rep Meth, PMID: 35880016).

Concerning the result that 405 nm signal changed in the same direction as 470 nm signal, we address this in detail above in response to a similar critique from Reviewer #1 and will reiterate here. In short, while the spectrum data suggest that we would see opposite signals at 405 nm and 470 nm (**Supplementary Fig 3a-b**), there are other factors *in vivo* that can contribute to the fluorescence intensity, such as pH (which we have now characterized extensively in **Supplementary Fig 4**), which could explain this discrepancy. Based on our new findings, one possibility explaining how the 405 and 470 channels could have signals in the same direction is that the behavior driving N/OFQ release may also cause the local pH to decrease, which would cause an increase in fluorescence at the 405 nm wavelength relative to its intensity at the higher pH (**Supplementary Fig 4c**). Furthermore, our new findings also indicate that this sensitivity to pH changes is more prominent at wavelengths closer to 405 nm than wavelengths closer to 435 nm, which could also explain why we observed smaller changes in the 435 nm channel than the 405 nm channel.

By and large, the small increase in NOPLight fluorescence emission observed *in vivo* when exciting at 405 nm upon RO-64 administration is thus most likely an artifact caused by the physiological response of the animal to the administration of the compound. Importantly, such artifact is much smaller in amplitude when compared to the real functional sensor signal recorded at the at 470 nm, which reflects the sensor's direct response to the agonist. However, we have ultimately removed Supplementary figures 6 and 7 to avoid this understandable confusion in lieu of stronger *in vivo* control experiments completed during revision using the NOPR antagonist J113397 as well as the control sensor NOPLight-ctr. We believe our new *in vivo* experiments address the concerns about whether our reported dynamics of endogenous N/OFQ release are real by demonstrating that the recorded NOPLight signal is blocked in the

presence of a NOPR antagonist (**Fig 5a-f**) and that the control sensor NOPLight-ctr has no response during the behavior that produced the largest NOPLight response.

2) Seems like interpretation in Supp Figure 6 and 7 are based on a single example mouse. Given this is major issue with the study, investigating the 405 and 435 signals across more than 1 mouse should be done to build any confidence in the results. In addition, all of the in vivo data is presented entirely qualitatively and it appears that no statistical quantification is performed.

We agree that it was unconvincing to report representative data and to not provide statistical analyses for the *in vivo* behavioral data. We completed additional control experiments (described above), and all behavioral fiber photometry recordings are now reported as mean \pm SEM. We have also completed rigorous statistical analysis across all behaviors to quantify the changes in NOPLight signal that we describe in the manuscript text.

3) The authors have nicely generated a control sensor that doesn't respond to ligand, but do not use this sensor to address artefact issues in the in vivo photometry recordings. This is important to both show the usefulness of using this control sensor and to strengthen any evidence that NOPLight can detect real, naturally released peptide.

We agree that it is important to show our control sensor NOPLight-ctr during *in vivo* recordings. As described above, we completed new experiments using the control sensor which demonstrate its lack of response during a freely-moving behavior and also further support our findings that NOPLight is detecting real, endogenous N/OFQ release *in vivo*.

4) Given comments #1 – 3, I am not convinced that this generation of NOPLight, as determined by the data presented in this manuscript, would be able to reliably detect natural dynamics in N/OFQ. However, the sensor does appear able to track chemogenetically-evoked release and agonists. As such, without adding further engineering of the sensor, I would suggest rewording the title and text to not overstate the ability of detecting natural N/OFQ dynamics and address this limitation in the discussion.

Given the findings presented in our original submission, we can understand why the Reviewer would make this recommendation. However, we have made extensive efforts to address any concerns about NOPLight's ability to detect endogenous N/OFQ release and believe that our manuscript's title and text are now strongly supported by our data. We demonstrated that NOPLight signal during sucrose consumption is blocked by NOPR antagonism despite animals making a similar number of licks in the presence of the antagonist, indicating that the signal we detected is dependent on NOPLight's ability to bind and detect N/OFQ. Furthermore, we completed this test while animals were head-fixed to dispel remaining concerns that our signal was a result of motion artefact. We also used our control sensor in a freely-moving behavior that produced the greatest NOPLight signal change we detected, finding that NOPLight-Ctr signal was remarkably stable throughout the behavior. Finally, our recordings of NOPLight signal across multiple Pavlovian and operant training sessions consistently demonstrate that NOPLight signal decreases during consumption of a reward obtained as a result of performing an operant behavior (**Fig 6, Supplementary Fig 9**). This signal is tightly aligned to the average consumption window for a given training day and is absent on trials where animals perform the operant behavior but do not obtain a reward (**Fig 6d, g, Supplementary Fig 9i**). We completed statistical quantifications of all NOPLight recordings *in vivo* which reliably support that NOPLight signal is significantly different during the tested behavioral events relative to baseline levels and/or to control signal (pre-treatment with J113397 or NOPLight-ctr). We hope that taken together, our new experiments and analyses provide substantial evidence that our sensor does detect natural N/OFQ dynamics.

5) The authors characterized NOPLight onset dynamics but did not report offset measurement which are important for the community. Similar fast removal of ligand and measurement of signal decay would address this.

We thank the Reviewer for raising the need for this important characterization of the sensor. We have performed additional experiments to address this. Please see our updated Figure 1g where we include NOPlight kinetics during ligand washout and local ligand application, as well as the new Supplementary Figure 6 measuring NOPlight OFF-kinetics in acute brain slices. Please also see our answer to Reviewer #1, who asked us for the same characterization.

6) There are some major issues with Results section pointing to the correct Supplementary Figure 7 and 8 panels. Supp Fig 7a,b are never mentioned in the results. Supp Figure 7 c,d, are toe pinches but results section refers to them as reward experiments. Supp Figure 8 is supposed to be aversive data but is all appetitive data.

We appreciate the Reviewer pointing out these inconsistencies. Supplementary Figures 7 and 8 have been replaced in this resubmission, so this concern should be addressed.

7) The figures legends need more clear description of what the sample size was as in current format it is left to the reader to guess. In legends for Figure 5, for example, is the mean and sem across mice and therefore the sample size is 4?

We carefully reviewed all figure legends and have now clarified the sample size for all experiments to the best of our knowledge.

Reviewer #3 (Remarks to the Author):

Thank you for sending me this interesting piece of Internationally collaborative work to review. The N/OFQ-NOP system has wide applicability across a number of fields and as such Nature Comms is an appropriate vehicle for dissemination. The authors indicate “The functional relevance of N/OFQ action in the mammalian brain remains unclear due to a lack of high-resolution approaches to detect this neuropeptide with appropriate spatial and temporal resolution” This study describes NOPlight – a genetically encoded sensor that reports the release of N/OFQ “with unprecedented temporal resolution ex vivo and in vivo”. My own group published a method to measure single cell release from isolated immune cells (DOI: 10.1371/journal.pone.0268868) but this is limited, cumbersome and can only be used in vitro. NOPlight is a paradigm shift by comparison with much wider applicability – critically for use in vivo and with behavioural correlates.

The paper is complex but well written and easy to follow. I have the following comments to make;

1. Potency values from log CRC would be better expressed at pEC50 values. N/OFQ potency at the sensor is high but a little weaker than in CNS (Ferrara paper – by reference).

We have added the corresponding pEC50 values in the manuscript's text.

2. It is interesting UFP-101 displayed no residual agonist activity – presumably expression was not huge ?

3. In vitro conclusions “These results indicate that while NOPlight retains the ligand selectivity of its parent receptor, its cellular expression has a very low likelihood of interfering with intracellular signaling; thus the sensor can be meaningfully utilized in physiological settings.” are supported by the data.

We would like to thank the Reviewer for his comments.

4. Fig 2c does not saturate and looks very weak. Without the top 2 concentrations does the rest of the curve saturate ?

We believe that in acute brain slices it is difficult to use bath application to reach an exact concentration of peptide at the surface of neurons in a homogeneous manner due to several well established factors in the history of neuropeptide research. This is because the slice thickness (usually 300-350 μm) poses a diffusion barrier to the peptide. In addition, it is difficult

to account for peptide degradation by peptidases expressed in tissue. These factors likely explain why application of higher concentrations of N/OFQ to the perfusion is not sufficient to reach saturation. We do not believe that it would be correct procedure to remove the top concentration datapoints and fit a dose-response curve to an incomplete dataset.

5. The system provides robust antagonist (LY and J) sensitive readout for the Roche agonist in vivo.

6. Chemogenetic activation of VTA neurons increased N/OFQ (measured with the biosensor) that was prevented by a NOP antagonist (LY).

7. Linking release to behaviour is one of the real strengths of this study and widens the appeal.

8. I have no comments on the structure of the figures.

In summary, this excellent piece of work with technical rigour and wide applicability receives by strongest support for publication. My comments should be viewed as very minor.

David Lambert, Leicester UK.

(For full disclosure – I have provided NO confidential comments to the editor)

We wish to thank the Reviewer for carefully reading through our manuscript and pointing out his comments above.

Reviewer #4 (Remarks to the Author):

Zhou, Stine, et al. describe the engineering and characterization of a fluorescent biosensor for the opioid peptide nociceptin called NOPLight. The sensor was constructed by insertion of circularly permuted GFP into intracellular loop 3 of the human nociceptin receptor (NOPR), analogous to previously described cpGFP-GPCR sensors. By testing ~100 variants within the linkers and intracellular loop 2 they optimized the max fluorescence response in cell culture to delf/F of ~4. The response time, ligand affinity, and membrane trafficking of the sensor appear reasonable and they demonstrate that the sensor does not contribute to GPCR-mediated intracellular signaling. They additionally make a "dead" sensor by introducing mutations that block ligand binding. The authors characterize the performance of NOPLight in cell culture, acute brain slice, and in vivo in mouse brain in response to added nociceptin, receptor agonists and antagonists, chemogenetic circuit activation, and naturalistic stimuli during Pavlovian conditioning.

Generally speaking, the sensor appears to perform reasonably, especially in vitro, and the results are well described and presented. This tool will likely be useful for the community and I would recommend publication, although the demonstration of in vivo responses to naturalistic stimuli is not super convincing.

My primary concerns are as follows:

-No amino acid sequence of the sensor is described in the manuscript (that I saw), this should be added

We would like to thank the Reviewer for noticing that. We now added a file to the submission called Supplementary Data S1 where both DNA and protein sequences of NOPLight can be found. We also highlighted the residues mutated to alanine in NOPLight-ctr therein.

-The in vivo responses (Fig 5) are quite small and noisy. They are likely real since there is a qualitative difference during the consumption phase between rewarded and non-rewarded trial, but these data are not terribly convincing. Indeed, the data did not appear to be completely reproducible when repeated using multiple wavelengths in the fiber photometry for normalization (supp fig 8). The "dead" sensor was not used in these experiments and could

be a good control. Are all of the data in Fig 5 from n=4 mice as suggested in caption 5a? If not this should be stated.

We agree that our *in vivo* data as originally presented was not convincing enough. Our resubmission thoroughly addresses this concern as we have discussed in detail above, but in short we completed several new *in vivo* experiments, added relevant controls using the NOPR antagonist J113397 and our control sensor, and included rigorous statistical analyses that we believe strengthen our findings and sufficiently address this concern.

REVIEWER COMMENTS

Reviewer #1 (Remarks to the Author):

The revised manuscript from Zhou et al. addresses some of the concerns raised in the previous review. However, there are two areas where further clarification and additional data could strengthen the paper.

pH Effect:

The authors focus on the effect of extracellular pH on NOPLight. While this is valuable, considering the intracellular location of the fluorescent module, it would be beneficial to investigate how intracellular pH might also influence the sensor's performance. Previous studies have shown that intracellular pH can change during neuronal activity. Including measurements of intracellular pH effects would provide a more comprehensive understanding of NOPLight's behavior.

NOPLight-ctrl for Figure 6:

Figure 6 presents a potentially very important finding, suggesting that NOPLight accurately reports endogenous NOP activity in living mice. This would be a significant contribution to the study of NOP biology in vivo. To further strengthen this result, it would be helpful to see data demonstrating that the observed signal is absent when using a mutant control sensor. This would provide even stronger evidence that NOPLight is faithfully reporting NOP dynamics.

Reviewer #2 (Remarks to the Author):

The additional in vivo studies using antagonists, the control sensor, and assays driving more robust changes have addressed my initial concerns about the detection of endogenous ligand. As such, I support publication of this revised manuscript. I congratulate the authors on this contribution to the field.

Reviewer #2 (Remarks on code availability):

I briefly reviewed the code. It is not in a format that would be easily used by the community without some instructions and effort. However, it is generally in line with the state of most code in the neuroscience field.

Reviewer #3 (Remarks to the Author):

Thank you for addressing my relatively minor comments. I liked the original and that has not changed. This is a strong MS that will appeal to wide readership. There are no specific comments for the 'Editors'

Reviewer #4 (Remarks to the Author):

The authors have sufficiently addressed my concerns from the previous review.

We would like to thank all the Reviewers for the positive feedback on our revised manuscript and the appreciation of the substantial efforts we put into this revision. With our answers below we would like to address the remaining concerns of Reviewer #1. The original Reviewer's comments are shown in *blue italics*. Cited text from the revised manuscript is shown here in *black italics*. We also marked in blue any major text changes in the revised version of our manuscript to enable transparency and ease of identification.

Reviewer #1 (Remarks to the Author):

The revised manuscript from Zhou et al. addresses some of the concerns raised in the previous review. However, there are two areas where further clarification and additional data could strengthen the paper.

pH Effect:

The authors focus on the effect of extracellular pH on NOPLight. While this is valuable, considering the intracellular location of the fluorescent module, it would be beneficial to investigate how intracellular pH might also influence the sensor's performance. Previous studies have shown that intracellular pH can change during neuronal activity. Including measurements of intracellular pH effects would provide a more comprehensive understanding of NOPLight's behavior.

While we recognize the importance of pH changes as a potential source of artefacts, we would like to point out that the suggested experiment (i.e. measuring intracellular pH) is technically very challenging for us to do. In order to do that we would need to establish in our hands the use of either pH-sensitive microelectrodes (PMID: 20419443), pH-sensitive dyes that require careful calibration (PMID: 19831417), or similar approaches. Currently, we are not set up to perform any of these measurements in-house in an accurate, physiologically-meaningful manner, and within a reasonable timeframe for this revision.

We would also like to note that pH alterations pose a much more important issue for sensors that are positioned the extracellular side of the cell membrane [e.g. sensors based on bacterial Periplasmic Binding Proteins (PBP), such as the glutamate sensor iGluSNFR (PMID: 23314171)]. These sensors can in some instances (e.g. when present nearby synaptic sites) be exposed to rapid pH fluctuations which occur locally during neurotransmitter release, due to vesicle fusion events. However, in GPCR-sensors the cpGFP is facing the intracellular side of the cell membrane. In contrast to the extracellular environment, cells evolved intrinsic mechanisms to maintain tight control over intracellular pH conditions, given the large impact that [H⁺] has on most biochemical reactions, cell metabolism and cell health, and this is particularly the case in neurons (PMID: 1847131). Under physiological conditions, neural activity-dependent pH changes in the intracellular environment of neural cells are known to be rather small in size (they were quantified to be within the 0.02-0.1 ΔpH range by PMID: 33033238 and PMID: 28768175). For these reasons we believe that neuronal activity-dependent intracellular pH fluctuations will not have a major impact on the fluorescence response of GPCR-sensors, including NOPLight. In addition, in order to accurately establish the p.H. sensitivity of an indicator one would need to exactly control the p.H. of the solution in which the protein is found, which can only be done on purified protein. While soluble sensors like GCaMPs or iGluSNFR can be purified and tested in such way, this is technically challenging to do in the case of GPCR-sensors and would certainly fall beyond our capacities and field of expertise. Likely for this reason, none of the previously published GPCR-based sensors has been characterized for p.H. sensitivity in this way.

In spite of these limitations, we did our best to attempt to experimentally address the concern raised by the Reviewer to the best of our abilities. The concern of the Reviewer stems from the fact that during neuronal activity the intracellular pH can slightly change (although

generally the intracellular pH falls within a stable and very narrow range, as mentioned above). Considering this, we believe that the most informative experiments to address this issue should be measurements of NOPLight's fluorescence performed simultaneously during neuronal activity changes. To experimentally address this remaining concern, we performed a new experiment in which we monitored the fluorescence response of NOPLight to N/OFQ in cultured neurons under unstimulated conditions and upon stimulation of neuronal firing activity evoked by glutamate (similar to previous work, for example see PMID: 12060807). In order to also monitor the firing activity of the neurons we co-expressed a red calcium indicator (JRCaMP1b) along with NOPLight, and performed simultaneous dual color imaging of the two indicators. Our findings, shown in the new Supplementary Figure 5, indicate that during periods of glutamate-evoked neural activity the fluorescence response of NOPLight to N/OFQ remained stable, indicating that changes to the intracellular environment occurring during neuronal activity (which include physiological intracellular p.H. fluctuations) do not cause fluctuations in NOPLight's fluorescent response. A description of the new results has been added to the results section of the manuscript as follows:

“Lastly, we evaluated whether changes in neural activity could cause alterations in the observed fluorescence of NOPLight. To do this we co-expressed NOPLight with the red genetically encoded calcium indicator JRCaMP1b in primary cultured neurons via viral transduction, followed by simultaneous multiplexed imaging of NOPLight fluorescence and neuronal calcium activity in the absence and presence of bath-applied glutamate at a concentration known to evoke neuronal activity (5 μ M). Overall we did not observe noticeable changes in the fluorescence of NOPLight during periods of evoked neuronal activity (Supplementary Figure 5), suggesting that potential alterations in the intracellular environment occurring during neuronal activity are not likely to influence the fluorescent responses of NOPLight”.

NOPLight-ctrl for Figure 6:

Figure 6 presents a potentially very important finding, suggesting that NOPLight accurately reports endogenous NOP activity in living mice. This would be a significant contribution to the study of NOP biology in vivo. To further strengthen this result, it would be helpful to see data demonstrating that the observed signal is absent when using a mutant control sensor. This would provide even stronger evidence that NOPLight is faithfully reporting NOP dynamics.

While we recognize that using the control sensor in this experiment would add additional strength, we feel the effort is not effectively practical nor the further use of animals ethically justifiable given that we show NOPR antagonism blocks NOPLight signal in an equivalent head-fixed behavior (Figure 5) as well as in multiple other contexts across chemogenetic, pharmacological, and in vitro experiments, and also demonstrated an absence of in vivo signal when using the control sensor (Figure 5). Additionally, all of our fiber photometry recordings are controlled for internally using an isosbestic channel to correct for motion artefact, which lends further strength to our results reflecting true dynamics. We have established within a rigorous manner our capacity to detect endogenous release with the sensor.

This sensor and its development represent step 1 of a process whereby we hope to further improve kinetics and sensitivity of the tool. This includes generating additional control sensors, testing the sensitivity of NOPLight across many other brain regions where there is likely more cellular N/OFQ expressing neurons, and those with less. We could iterate this for two more years, testing more controls, mice, and assays, but the better approach is to give the community additional opportunities to characterize the tool independently from our groups, while we also work simultaneously to further advance this biosensor's abilities.

Finally, the use of animals at our institution and by NIH compliance standards requires we have good faith rationale for using animals to conduct additional experiments. In this request,

we believe strongly that while some new information could come to light, it is not ethically or financially responsible (during a very challenging time for NIH funded researchers) to use additional animals for a control that is essentially established in a prior figure. We feel that throughout our in vivo studies we demonstrated alternate approaches in controlling for NOPLight detection of N/OFQ dynamics using NOPR antagonism and the mutant sensor across different experiments while being mindful of the 3Rs of animal research and considerate of whether use of additional animals was justified.

We sincerely appreciate the reviewers comments, and have done our best to address concerns. We also added a comment in the text, which makes it clear more controls and testing are needed with the sensor, as follows:

“Our work here lays critical groundwork that will need to be built upon through continued use of NOPLight and its control variant in similar reward-related behaviors, different brain regions, and with additional controls, which will be important for optimizing their implementation and developing general best practices amidst the rapidly expanding use of fluorescent peptide sensors.”

Reviewer #2 (Remarks to the Author):

The additional in vivo studies using antagonists, the control sensor, and assays driving more robust changes have addressed my initial concerns about the detection of endogenous ligand. As such, I support publication of this revised manuscript. I congratulate the authors on this contribution to the field.

Reviewer #2 (Remarks on code availability):

I briefly reviewed the code. It is not in a format that would be easily used by the community without some instructions and effort. However, it is generally in line with the state of most code in the neuroscience field.

Reviewer #3 (Remarks to the Author):

Thank you for addressing my relatively minor comments. I liked the original and that has not changed. This is a strong MS that will appeal to wide readership. There are no specific comments for the 'Editors'

Reviewer #4 (Remarks to the Author):

The authors have sufficiently addressed my concerns from the previous review.

REVIEWERS' COMMENTS

Reviewer #1 (Remarks to the Author):

Thanks for considering my feedback on the discussion section. While I initially had some reservations, I believe that including it would be very helpful for readers.